# ARLON: Boosting Diffusion Transformers with Autoregressive Models for Long Video Generation

**Zongyi Li**[1*]**, Shujie Hu**[2*]**, Shujie Liu**[3†]**, Long Zhou**[3]**, Jeongsoo Choi**[4]**, Lingwei Meng**[2]**,
**Xun Guo**[3]**, Jinyu Li**[3]**, Hefei Ling**[1]**, Furu Wei**[3]

[1] Huazhong University of Science and Technology
[2] The Chinese University of Hong Kong
[3] Microsoft Corporation
[4] KAIST

## Abstract

Text-to-video (T2V) models have recently undergone rapid and substantial advancements. Nevertheless, due to limitations in data and computational resources, achieving efficient generation of long videos with rich motion dynamics remains a significant challenge. To generate high-quality, dynamic, and temporally consistent long videos, this paper presents ARLON, a novel framework that boosts diffusion Transformers with autoregressive (**AR**) models for long (**LON**) video generation, by integrating the coarse spatial and long-range temporal information provided by the AR model to guide the DiT model effectively. Specifically, ARLON incorporates several key innovations: 1) A latent Vector Quantized Variational Autoencoder (VQ-VAE) compresses the input latent space of the DiT model into compact and highly quantized visual tokens, bridging the AR and DiT models and balancing the learning complexity and information density; 2) An adaptive norm-based semantic injection module integrates the coarse discrete visual units from the AR model into the DiT model, ensuring effective guidance during video generation; 3) To enhance the tolerance capability of noise introduced from the AR inference, the DiT model is trained with coarser visual latent tokens incorporated with an uncertainty sampling module. Experimental results demonstrate that ARLON significantly outperforms the baseline OpenSora-V1.2 on eight out of eleven metrics selected from VBench, with notable improvements in dynamic degree and aesthetic quality, while delivering competitive results on the remaining three and simultaneously accelerating the generation process. In addition, ARLON achieves state-of-the-art performance in long video generation, outperforming other open-source models in this domain. Detailed analyses of the improvements in inference efficiency are presented, alongside a practical application that demonstrates the generation of long videos using progressive text prompts. Project page: `http://aka.ms/arlon`.

## 1 Introduction

Text-to-video (T2V) models have recently undergone rapid advancements, driven by both Transformer architectures (Vaswani, 2017) and diffusion models (Ho et al., 2020). Autoregressive (AR) models, such as decoder-only Transformers, offer notable advantages in scalability and long-range in-context learning (Wang et al., 2023a), demonstrating strong potential for video generation from text (Yan et al., 2021; Ge et al., 2022; Hong et al., 2022; Yu et al., 2023; Kondratyuk et al., 2023). Meanwhile, diffusion-based models, including U-Net and Diffusion Transformers (DiT), have set a new benchmark in high-quality video generation, establishing themselves as dominant approaches in the field (Zheng et al., 2024; Lab & etc., 2024). However, despite the rapid progress in DiT models, several key challenges remain: 1) High training cost, especially for high-resolution videos,

---

[*]Equal contribution. Work was done during internship at Microsoft Research Asia.
[†]Corresponding author.

resulting in insufficient **motion dynamics** as training is restricted to short video segments within each batch; and 2) The inherent **complexity and time-consuming** nature of generating videos entirely through denoising based solely on text conditions; and 3) Difficulty in generating long videos with **consistent motion and diverse content**.

Previous research typically employed autoregressive approaches for long video generation with DiT models (Weng et al., 2024; Gao et al., 2024; Henschel et al., 2024), generating successive video segments conditioned on the last frames of the previous segment. However, computational constraints restrict the length of these conditioned segments, resulting in limited historical context for the generation of each new segment. Additionally, when given the same text prompts, generated short video segments often feature identical content, increasing the risk of repetition throughout the entire overall long video. Diffusion models, while excellent at producing high-quality videos, struggle to capture long-range dependencies and tend to generate less dynamic motion. Additionally, the requirement for multiple denoising steps makes the process both computationally expensive and slow. By contrast, AR models are more effective at maintaining **semantic information** over long sequences, such as motion consistency and subject identity. They excel at generating continuous actions without the repetition issues that diffusion models often encounter in long video generation, and the inference of AR models typically achieves faster inference times.

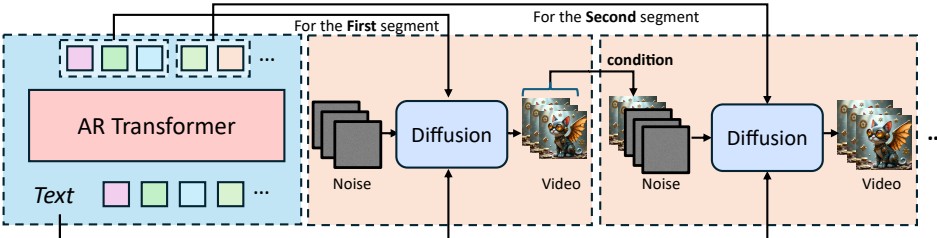

Figure 1: Generation process for long videos with autoregressive transformer and DiT.

Leveraging AR for coarse predictions, we can enhance the diffusion model's capacity to capture richer dynamics and maintain continuity in long video sequences. The AR-generated features can also serve as an initialization to accelerate the diffusion process, resulting in faster and more efficient generation of long-form videos. Building on this concept, we propose ARLON, a novel framework that effectively combines the advantages of autoregressive Transformer and DiT models for long video generation. As shown in Fig. 1, ARLON first generates long-term, coarse-grained discrete visual units (AR codes) autoregressively using a decoder-only Transformer. These discrete AR codes are then segmented and sequentially fed into the DiT model by the proposed semantic injection module, which autoregressively generates high-quality video segments. Specifically, the first N seconds of AR codes guide the DiT model to generate the first video segment as illustrated in the middle part of Fig. 1. The second N second of AR codes, along with the last M seconds of the first video segment, serve as the condition to generate the subsequent video segment.

As shown in Figure 2, to bridge the feature spaces between the AR Transformer and DiT model, as well as to balance the learning complexity of the AR model and the information density of the visual tokens, we employ a 3D latent VQ-VAE to compact and quantize the input latent features of the DiT model into discrete tokens. Various architectures of the semantic injection module, such as MLP adapter, adaptive norm modules, and ControlNet, are explored to ensure the coarse-grained AR codes guide the DiT model effectively. However, unlike scenarios where conditioned images, videos, or motion trajectories (Chen et al., 2023b; Zhang et al., 2024; Peng et al., 2024) are available, the tokens generated by autoregressive models in text-based video generation scenarios tend to be noisy, leading to a noticeable drop in accuracy when transitioning from teacher-forcing training to autoregressive inference. To address this, we propose two key innovations: 1) training the DiT model using coarse visual latent tokens generated by a different latent VQ-VAE with higher compression rate than those used for AR model training, and 2) integrating an uncertainty sampling module into the semantic injection module to further enhance model performance.

Our ARLON model is evaluated using the VBench (Huang et al., 2024) video generation benchmark. Experimental results demonstrate that ARLON outperforms the baseline OpenSora-V1.2 on eight out of eleven metrics, while also delivering competitive performance across other metrics. Additionally, ARLON achieves state-of-the-art performance by effectively generating high-quality,

temporally coherent, and dynamically rich long videos, surpassing other open-source models in this area. Our contributions can be summarized in three points:

- We present ARLON, a novel framework that seamlessly combines the strengths of autoregressive Transformers and Diffusion Transformers (DiT). In this approach, the AR model supplies coarse spatial and long-range temporal information, effectively guiding the DiT model to generate long, high-quality videos with rich dynamic motion.

- To bridge the AR and DiT models while balancing learning complexity and information density, a latent VQ-VAE is introduced to compress the DiT model's input space into compact, highly quantized visual tokens. These tokens are then used to train an autoregressive Transformer model, generating visual tokens based on the input text prompt. To reduce the noise inevitably introduced during AR inference, we introduce two noise-resilient strategies for the DiT model training: coarser visual latent tokens and uncertainty sampling.

- Both quantitative and qualitative analyses of the improvement in inference efficiency are given. In addition, long video generation using progressive text prompts is implemented, where each subsequent prompt builds on the previous.

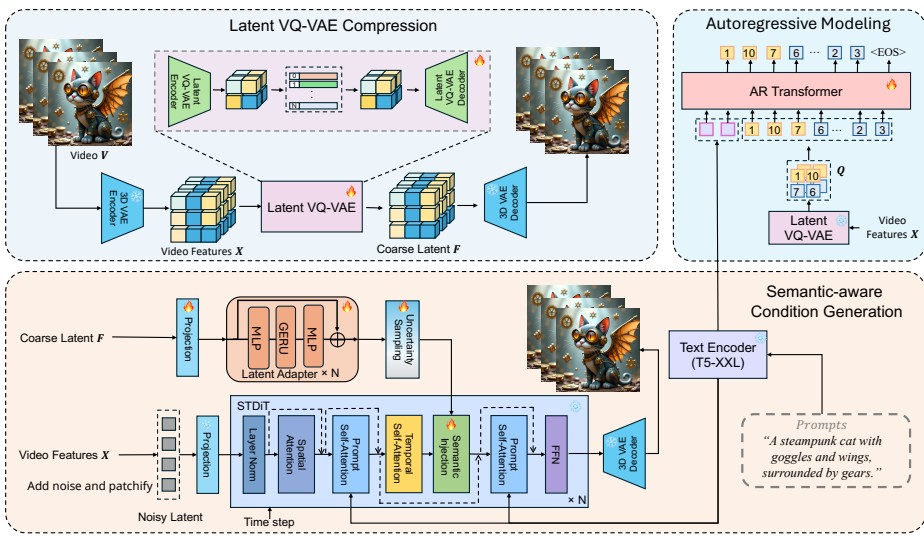

Figure 2: Overview of the ARLON framework, which consists of three key components: Latent VQ-VAE Compression, Autoregressive Modeling, and Semantic-aware Condition Generation.

## 2 METHOD

As illustrated in Figure 2, our ARLON comprises three primary components: Latent VQ-VAE Compression, Autoregressive Modeling, and Semantic-aware Condition Generation. Given a text prompt, the autoregressive (AR) model predicts coarse visual latent tokens, which are constructed from a 3D VAE encoder followed by a latent VQ-VAE encoder based on the target video. These predicted visual latent tokens encapsulate both the coarse spatial information and consistent semantic information. Based on these tokens, a latent VQ-VAE decoder generates continuous latent features, which serve as semantic conditions to guide the DiT model with a semantic injection module. To mitigate the noise inevitably introduced during AR inference, we introduce two noise-robust training strategies: 1) coarser visual latent tokens, and 2) uncertainty sampling module.

### 2.1 AUTOREGRESSIVE

**Latent VQ-VAE** To align the feature spaces between the AR and DiT, and obtain quantized and compact discrete visual tokens, a latent VQ-VAE nested within the 3D VAE of the DiT model is constructed. Following Yan et al. (2021), the latent VQ-VAE encoder $E_{latent}$ consists of 3D convolutional neural network (CNN) blocks and residual attention blocks, followed by a decoder $D_{latent}$ structured as the reverse of the encoder. Let the inputs of the VQ-VAE be denoted as $X \in \mathbb{R}^{T \times H \times W \times C}$. If the spatial and temporal compression factors of the 3D CNN encoder are $r$ and $o$ respectively, the encoder produces latent embeddings $V \in \mathbb{R}^{\frac{T}{r} \times \frac{H}{o} \times \frac{W}{o} \times h}$. Each embedding vector

$v \in \mathbb{R}^h$ in $V$ is quantized to the closest entry $c \in \mathbb{R}^m$ in the learned codebook $C \in \mathbb{R}^{K \times m}$. The index of each entry $c$ is used to represent the latent embeddings $V$ as $Q = \{1, 2, .., K\}^{\frac{T}{r} \times \frac{H}{o} \times \frac{W}{o}}$. For decoding, given the indices of video tokens, we can retrieve the corresponding entry $c$, which is used to obtain reconstructed video embeddings $F$ using the latent VQ-VAE decoder.

**Autoregressive Modeling** We employ a causal Transformer decoder as the language model to autoregressively generate discrete visual tokens from textual input. Specifically, the indices of visual tokens $Q$ are subsequently decomposed into 1D spacetime patches $Q^{AR} = [q_1, q_2, ..., q_N]$, with <EOS> appended in the end of whole video, and <FRAME> inserted at the end of each frame, where $N = (\frac{T}{r} + 1) \times \frac{H}{o} \times \frac{W}{o}$ and $q_i \in \{1, 2, ..., K\}$. These indices are converted into embeddings by the video code embedding layer, added with a learnable position embedding. The text based condition $Y$ is the contextual embedding of the video captions, generated by the T5 encoder (Raffel et al., 2020). The AR model, comprising blocks of multi-head attention and feed-forward layers, takes the concatenation of text and visual embeddings as input to model the dependency between these information, and the model is optimized to maximize the following probability

$$p(Q^{AR}|Y; \Theta_{AR}) = \prod_{n=1}^{N} p(q_n|Y, Q^{AR}_{<n}; \Theta_{AR}). \tag{1}$$

## 2.2 SEMANTIC-AWARE CONDITION GENERATION

**STDiT** Our ARLON framework is built on a spatial-temporal Transformer (Zheng et al., 2024), which serves as the backbone model. Given an input image latent $z$, a 3D embedding layer first projects the image into non-overlapping patches, which are then flattened. These flattened features are subsequently augmented with spatial and temporal position embeddings using ROPE (Su et al., 2024). The augmented features are then processed through a series of spatial-temporal DiT blocks, which conclude with an unpatchify layer predicting the noise. Each spatial-temporal DiT block comprises sequential modules for spatial-attention, temporal-attention, and cross-attention:

$$\begin{aligned} \mathbf{X} &= \mathbf{X} + \text{SpatialAttn}(\text{LN}(\mathbf{X})), \\ \mathbf{X} &= \text{rearrange}(\mathbf{X}, (bt)sd \rightarrow (bs)td), \\ \mathbf{X} &= \mathbf{X} + \text{TempAttn}(\text{LN}(\mathbf{X})), \end{aligned} \tag{2}$$

where $(bt)sd \rightarrow (bs)td$ means rearranging the tensor by merging the batch size $b$ with the spatial dimension $s$ and isolating the temporal dimension $t$ for subsequent temporal attention processing. The text information is incorporated with DiT features using cross-attention, providing auxiliary information to enhance spatial-temporal consistency in video generation.

**AR Semantic Condition** Videos can be compressed into a coarse latent space using a video VAE and latent VQ-VAE. As the AR model predicts tokens within latent VQ-VAE space, we leverage the reconstructed latent features from the latent VQ-VAE decoder as semantic conditions for training the diffusion model. These conditional features are subsequently employed to determine the coarse spatial and temporal content in the video. Given the video $x$, the corresponding conditional features $F$ can be extracted as:

$$F = D_{latent}(E_{latent}(E_{video}(x))). \tag{3}$$

Following the STDiT model, we initially project the coarse latent feature $F$ with a 3D embedding layer to generate the input condition $F_0$, followed with several adapter layers to inject the semantic information into the video generation process. As shown in Figure2, the adapter block contains several residential MLP block:

$$F_{i+1} = \text{Adapter}_i(\text{LayerNorm}(F_i)) + F_i. \tag{4}$$

**Semantic Injection** To incorporate coarse semantic information into video generation, we inject the AR semantic condition into the DiT model to guide the diffusion process. Rather than directly adding the condition to each block, we utilize a gated adaptive normalization mechanism for condition injection. As shown in the upper part of Figure 3, the input latent variable $X_i$ is first processed with a layer normalization, and the conditional latent variable $F_i$ is processed with an uncertainty sampling (will be introduced in Section 2.3) to get $\hat{F}_i$, which is projected into three parameters: scale $\gamma_i$, shift $\beta_i$, and gated parameter $\alpha_i$, followed by the application of adaptive layer normalization to inject the conditioning information into the original latent variable. Additionally, to regulate

the latent strength, we introduce an extra gated layer, initialized to zero, to adaptively control the injected features by adding them to the original DiT feature $\boldsymbol{X_i}$:

$$\boldsymbol{\alpha}_i, \boldsymbol{\beta}_i, \boldsymbol{\gamma}_i = \text{MLP}\left(\hat{\boldsymbol{F}}_i\right),$$

$$\text{Fusion}(\boldsymbol{X}_i, \boldsymbol{\alpha}_i, \boldsymbol{\beta}_i, \boldsymbol{\gamma}_i) = \boldsymbol{\alpha}_i \odot \text{MLP}\left(\boldsymbol{\gamma}_i \odot \text{LayerNorm}(\boldsymbol{X_i}) + \boldsymbol{\beta}_i\right) + \boldsymbol{X_i}. \tag{5}$$

## 2.3 TRAINING STRATEGY

In the training phase, each training sample consists of three inputs: the original video, a textual prompt $\boldsymbol{Y}$, and the AR semantic condition $\boldsymbol{F}$. For each video, we first convert it into the latent space $\boldsymbol{X}^0$. Subsequently, a timestep $t$ is randomly sampled from the interval $[0, T]$, and noise is added to the video latent $\boldsymbol{X}^0$, resulting in $X^t$. Our ARLON is then optimized using the following procedure:

$$\mathcal{L} = \mathbb{E}_{\boldsymbol{X}^0, t, \boldsymbol{F}, \epsilon \sim \mathcal{N}(0,1)}\left[\left\|\epsilon - \epsilon_\theta\left(\boldsymbol{X}^t, t, \boldsymbol{Y}, \boldsymbol{F}\right)\right\|_2^2\right]. \tag{6}$$

To enable overlapping long video generation and image-to-video generation, we randomly unmask frames, leaving them noise-free to serve as conditioning frames. To tolerate the errors inevitably introduced during AR inference, we implement two noise-resilient training strategies: coarser visual latent tokens and uncertainty sampling.

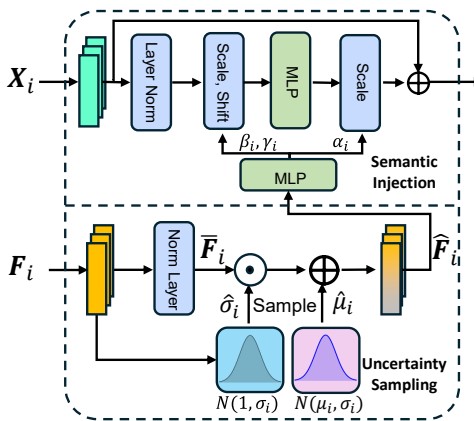

Figure 3: Semantic injection and uncertainty sampling.

**Coarser Visual Latent Tokens** During the training phase, we employ two variants of latent VQ-VAE with distinct compression ratios to enhance the diffusion process to tolerate noisy AR prediction results. Specifically, for AR training, our latent VAE utilizes a compression ratio of $(r, o, o)$, whereas for the DiT model training, we adopt a compression rate of $(r, 2o, 2o)$. By using noisier and coarser information in the model training, DiT model is encouraged to capture more general patterns, thereby reducing the risk of specific errors inevitably introduced by AR prediction results.

**Uncertainty Sampling** To simulate the autoregressive (AR) prediction variance, we introduce an uncertainty sampling module. As depicted in the lower part of Figure 3, this mechanism generates noise through the uncertainty sampling module rather than strictly relying on the original coarse latent features $\boldsymbol{F}_i$, and it is applied randomly after each adapter block during model training. Instead of injecting standard Gaussian noise, the noise is drawn from the original distribution of the latent features $\boldsymbol{F}_i$. The mean $\boldsymbol{\mu}_i$ and standard deviation $\boldsymbol{\sigma}_i$ of the noise are first calculated from the original coarse latent feature $\boldsymbol{F}_i$, and a normalization layer is applied to produce the whitened feature $\bar{\boldsymbol{F}}_i = \frac{\boldsymbol{F}_i - \boldsymbol{\mu}_i}{\boldsymbol{\sigma}_i}$. Following Chen et al. (2022), the sampled feature $\hat{\boldsymbol{F}}_i$ is calculated as:

$$\hat{\boldsymbol{F}}_i = \hat{\boldsymbol{\sigma}}_i \odot \bar{\boldsymbol{F}}_i + \hat{\boldsymbol{\mu}}_i, \quad \hat{\boldsymbol{\sigma}}_i \sim N\left(\boldsymbol{1}, \boldsymbol{\sigma}_i\right), \quad \hat{\boldsymbol{\mu}}_i \sim N\left(\boldsymbol{\mu}_i, \boldsymbol{\sigma}_i\right), \tag{7}$$

where $\hat{\boldsymbol{\sigma}}_i$ and $\hat{\boldsymbol{\mu}}_i$ represent noisy vectors sampled from the modeled mean and variance distribution of the target feature. To ensure that the sampled feature distribution closely approximates the original, the mean value of $\hat{\boldsymbol{\sigma}}_i$ distribution is set to 1.

## 3 RELATED WORK

### 3.1 TEXT-TO-VIDEO GENERATION

In recent years, substantial research has been dedicated to the development of text-to-video generation (T2V) models (Ho et al., 2022b;a; Singer et al., 2022; Chen et al., 2023a; Zhou et al., 2022; Wang et al., 2024; 2023b; Blattmann et al., 2023; Guo et al., 2023; Zeng et al., 2024; Ma et al., 2024; Peebles & Xie, 2023; Zheng et al., 2024; Lab & etc., 2024; Yang et al., 2024; Ju et al., 2024). These efforts can be broadly categorized into two main types: language-model-based and diffusion-model-based methods. For diffusion-model-based approaches, pioneering works such as VDM (Ho et al., 2022b) employ a 3D U-Net diffusion model for video generation. Imagen Video (Ho et al.,

2022a) and Make-a-Video (Singer et al., 2022) introduce spatiotemporally factorized models to generate high-definition videos. Subsequently, VideoCraft (Chen et al., 2023a) and Magic Video (Zhou et al., 2022) utilize Video VAE and larger datasets to enhance the generalization capabilities of video models. Magic Video V2 (Wang et al., 2024) and Lavie (Wang et al., 2023b) propose cascaded models for high-quality and aesthetically pleasing video generation. Moreover, SVD (Blattmann et al., 2023), Animatediff (Guo et al., 2023), and PixelDance (Zeng et al., 2024) employ T2I models to generate images and subsequently animate them into videos. Meanwhile, Latte (Ma et al., 2024) explores the training efficiency of video generation using a DiT model (Peebles & Xie, 2023), and SORA accelerates the investigation of DiT models. Recently, more DiT-based diffusion models have emerged, including OpenSora (Zheng et al., 2024), OpenSoraPlan (Lab & etc., 2024), CogvideoX (Yang et al., 2024), and Mira (Ju et al., 2024). While diffusion-based methods can generate high-quality videos, they are typically trained on fixed-length short videos (e.g., 16 frames), which limits their ability to produce longer videos. Furthermore, these methods predominantly focus on videos with small dynamic ranges. Language-model-based methods leverage the transformer architecture to predict the next latent code of video representations in an autoregressive manner. VideoGPT (Yan et al., 2021) and TATS (Ge et al., 2022) utilize GPT-like transformer models to generate extended video sequences. CogVideo (Hong et al., 2022) employs a transformer to produce key frames, followed by a second upsampling stage to achieve higher frame rates. Recently, Magvit2 (Yu et al., 2023) introduced a novel lookup-free quantization approach, enhancing visual quality in language-based models. VideoPoet (Kondratyuk et al., 2023) incorporates a mixture of multimodal inputs into large language models (LLMs) for synthesizing video tokens. Although transformer-based models can effectively capture long-range dependencies, they demand substantial resources for training long videos, and their quality still requires improvement.

## 3.2 Long Video Generation

The generation of long videos presents significant challenges due to inherent temporal complexity and resource constraints. Previous autoregressive GAN-based models (Ge et al., 2022; Yu et al., 2022; Skorokhodov et al., 2022) utilize sliding-window attention mechanisms to facilitate the generation of longer videos. However, despite these advantages, ensuring the quality of the generated videos remains problematic. Phenaki (Villegas et al., 2022) proposes a model for realistic video synthesis from textual prompts using a novel video representation with causal attention for variable-length videos, enabling the generation of arbitrary long videos. NUWA-XL (Yin et al., 2023) introduces a Diffusion over Diffusion architecture for extremely long video generation, allowing parallel generation with a "coarse-to-fine" process to reduce the training-inference gap. Recent diffusion-based models (Ma et al., 2024; Zheng et al., 2024) typically employ conditional mask inputs for overlapping generation. Although these mask generation methods can produce long videos, they often encounter issues related to temporal inconsistency. Recent advancements, such as StreamingT2V (Henschel et al., 2024), have introduced the injection of key frames into diffusion processes to enhance temporal consistency across different video segments. Additionally, some training-free approaches (Qiu et al., 2023; Lu et al., 2024) leverage noise rescheduling techniques to improve temporal consistency. Moreover, VideoTetris (Tian et al., 2024) presents a compositional framework for video generation. Our proposed method effectively integrates an autoregressive model for long-term coherence with a diffusion-based DiT model for short-term continuity, overcoming the limitations of existing techniques such as sliding window and diffusion-over-diffusion methods. This approach ensures video integrity and detail coherence over extended periods without repetition.

## 4 Experiments

### 4.1 Experimental Setup

**Dataset.** For the training of the latent VAE and AR transformer model, about 5.7M video clips are used, consisting of Openvid-1M (Nan et al., 2024), ChronoMagic-ProH (Yuan et al., 2024) and OpenSora-plan (including Mixkit, Pexels and Pixabay) (Lab & etc., 2024). For training the DiT model, we use 0.7M video clips from OpenVidHD-0.4M and Mixkit. To evaluate the performance of text-to-video generation, we utilize prompts from the Vbench benchmark (Huang et al., 2024) for comparison against other state-of-the-art models. In the ablation studies, 100 prompts are randomly selected from OpenVid-1M, excluded from the training dataset.

**Evaluation Metrics.** To assess the text-to-video generation, we employ evaluation metrics consistent with those used in VBench and Vbench-Long: Dynamic Degree, Aesthetic Quality, Imaging Quality, Subject Consistency, Background Consistency, Motion Smoothness, Temporal Flickering,

Temporal Style, Overall Consistency, Scene and Object Class. The first six metrics are used in the ablation studies.

**Implementation Details.** The time-space compression ratio of the latent VAE is 4×8×8 and 4×16×16 for the training of the AR model and DiT model, and the dimension and vocabulary size of the codebook are 256 and 2048 respectively. The AR model has the transformer structure with 12 layers, 16 attention heads, an embedding dimension of 1024, a feed-forward layer dimension of 4096, and a dropout of 0.1. DiT is initialized with OpenSora-V1.2, fixed during model training. The uncertainty sampling is employed randomly with a probability of 0.1. We use the Adam optimizer with a learning rate of $2 \times 10^{-5}$ for fine-tuning. The model is trained at a resolution of $512 \times 512$ and with a frame range from 51 to 136.

Table 1: Long video generation (600 frames) results of ARLON and other models on VBench. The higher scores of metrics indicate better performance.

| Models | Subject Consist | Background Consist | Motion Smooth | Dynamic Degree | Aesthetic Quality | Imaging Quality | Overall Consist |
|---|---|---|---|---|---|---|---|
| FreeNoise | 96.59 | 97.48 | 98.36 | 17.44 | 47.39 | **63.88** | 25.78 |
| StreamingT2V | 87.31 | 94.64 | 93.83 | **85.64** | 44.57 | 53.64 | 23.65 |
| OpenSora-V1.2 | 96.30 | 97.39 | **98.94** | 44.79 | 56.68 | 51.64 | 26.36 |
| **ARLON (Ours)** | **97.11** | **97.56** | 98.50 | 50.42 | **56.85** | 53.85 | **26.55** |

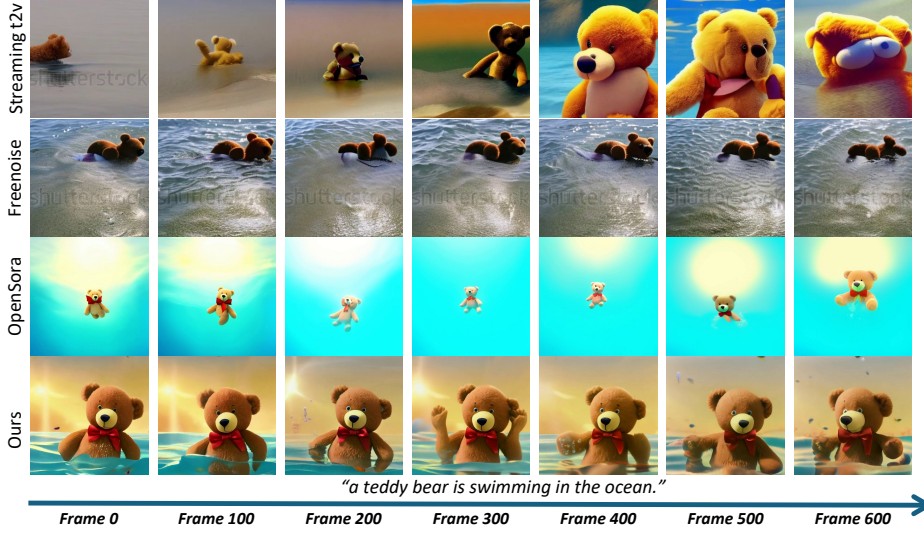

Figure 4: Qualitative comparisons between StreamingT2V, FreeNoise, OpenSora, and ARLON. Each video contains 600 frames.

## 4.2 RESULTS AND DISCUSSIONS

**Long Video Generation** We compare our ARLON model with other open-source text-to-long video generation models: StreamingT2V (Henschel et al., 2024), FreeNoise (Qiu et al., 2023), and OpenSora-V1.2 (Zheng et al., 2024). As shown in Table 1, while FreeNoise achieves the highest image quality, its low motion metrics indicate that the generated videos are mostly static or contain minimal movement. In contrast, StreamingT2V exhibits high levels of dynamism, but this comes at the cost of consistency and overall video quality. As the length of the generated video increases, the temporal coherence between different segments diminishes. Compared to the baseline model, OpenSora-V1.2, ARLON demonstrates significant improvements across almost all metrics. While the increase in dynamism leads to a slight reduction in motion smoothness, we consider this trade-off acceptable for the more dynamic and engaging video content ARLON produces. We also present several long video examples in Figure 4, while FreeNoise generates almost static videos, with minimal movement, such as the bear remaining stationary throughout. In contrast, our ARLON strikes a better balance, generating videos that not only exhibit dynamic motion but also maintain a high level of temporal consistency and natural flow.

Table 2: Performance comparison of Text-to-video (T2V) generation between our ARLON and other open-source or commercial models on VBench benchmark. The higher scores of metrics indicate better performance. The highlighted number in the top right corner reflects the improvements we achieved in comparison to OpenSora-V1.2.

| Models | Dynamic Degree | Aesthetic Quality | Imaging Quality | Subject Consist | Background Consist | Motion Smooth | Temporal Flicker | Temporal Style | Overall Consist | Scene | Object |
|---|---|---|---|---|---|---|---|---|---|---|---|
| Kling | 46.9 | 61.2 | 65.6 | **98.3** | 97.6 | 99.4 | 99.3 | 24.2 | 26.4 | 50.9 | 87.2 |
| Gen-2 | 18.9 | **67.0** | **67.4** | 97.6 | 97.6 | **99.6** | 99.6 | 24.1 | 26.2 | 48.9 | 90.9 |
| Pika-V1.0 | 47.5 | 62.0 | 61.9 | 96.9 | 97.4 | 99.5 | **99.7** | 24.2 | 25.9 | 49.8 | 88.7 |
| VideoCrafter-2.0 | 42.5 | 63.1 | 67.2 | 96.8 | **98.2** | 97.7 | 98.4 | 25.8 | **28.2** | **55.3** | 92.6 |
| LaVie | 49.7 | 54.9 | 61.9 | 91.4 | 97.5 | 96.4 | 98.3 | **25.9** | 26.4 | 52.7 | 91.8 |
| LaVie-Interpolation | 46.1 | 54.0 | 59.8 | 92.0 | 97.3 | 97.8 | 98.8 | 26.0 | 26.4 | 52.6 | 90.7 |
| Show-1 | 44.4 | 57.4 | 58.7 | 95.5 | 98.0 | 98.2 | 99.1 | 25.2 | 27.5 | 47.0 | **93.1** |
| CogVideo | 42.4 | 38.2 | 41.0 | 92.2 | 96.2 | 96.5 | 97.6 | 7.8 | 7.7 | 28.2 | 73.4 |
| OpenSoraPlan-V1.1 | 47.7 | 56.9 | 62.3 | 95.7 | 96.7 | 98.3 | 99.0 | 23.9 | 26.5 | 27.1 | 76.3 |
| OpenSora-V1.2 | 47.2 | 56.2 | 60.9 | 94.5 | 97.9 | 98.2 | 99.5 | 24.6 | 27.1 | 42.5 | 83.4 |
| **ARLON (Ours)** | **52.8**$^{\uparrow 5.6}$ | 61.0$^{\uparrow 4.8}$ | 61.0$^{\uparrow 0.1}$ | 93.4$^{\downarrow 1.1}$ | 97.1$^{\downarrow 0.8}$ | 98.9$^{\uparrow 0.7}$ | 99.4$^{\downarrow 0.1}$ | 25.3$^{\uparrow 0.7}$ | 27.3$^{\uparrow 0.2}$ | 54.4$^{\uparrow 11.9}$ | 89.8$^{\uparrow 6.4}$ |

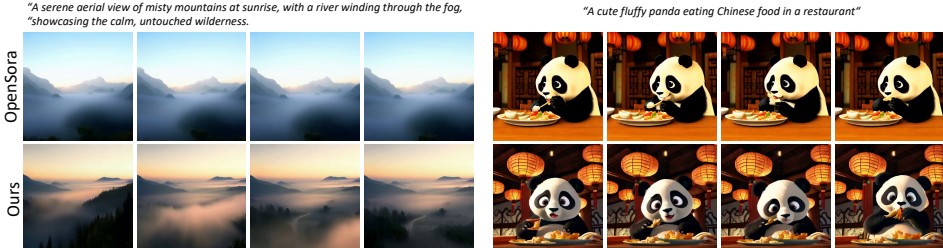

"A serene aerial view of misty mountains at sunrise, with a river winding through the fog, "showcasing the calm, untouched wilderness.

"A cute fluffy panda eating Chinese food in a restaurant"

Figure 5: Comparison of qualitative results for text-to-video generation.

**Text-to-video Generation** Our ARLON model is compared with other state-of-the-art open-source and commercial closed-source text-to-video generation models on VBench, including Kling[1], Gen-2[2], Pika-V1.0[3], VideoCrafter-2.0 (Chen et al., 2024), LaVie (Wang et al., 2023b), LaVie-Interpolation (Wang et al., 2023b), Show-1 (Zhang et al., 2023), CogVideo (Hong et al., 2022), OpenSoraPlan-V1.1 (Lab & etc., 2024) and OpenSora (Zheng et al., 2024). As shown in Table 2, we conducted a quantitative comparison between ARLON and ten other text-to-video models, where ARLON demonstrates superior performance in terms of dynamic degree while maintaining strong results across other metrics. Compared to the baseline OpenSora-V1.2, ARLON outperforms eight out of eleven metrics, with particularly significant improvements in dynamic degree, aesthetic quality, scene and object metrics, while achieving competitive results in the remaining three. Figure 5 shows the qualitative comparisons between ARLON and OpenSora. The scenes from OpenSora are mostly static, while ours exhibits significant camera movement and aligns better with the text.

## 4.3 ABLATION STUDY

The highly compact and quantized tokens generated by autoregressive (AR) models can introduce noise and errors when transitioning from teacher-forcing training to autoregressive inference. Effectively handling those noise and errors, several key factors are explored in the DiT model training: 1) the design of the semantic injection module, including its architecture and the number of DiT layers where it is applied; and 2) the training strategies employed for the DiT model to enhance its robustness. The results in Table 3 reveal several key points:
• Employing much coarser granularity during the DiT model training phase, specifically, utilizing the latent VAE with a compression ratio of $4 \times 16 \times 16$, compared to a ratio of $4 \times 8 \times 8$ during inference, can generate more inaccurate and noisier visual latent representations, which could make the DiT model tolerate the errors, thereby improving its robustness, and maintaining the consistency and qualities of the generated videos.
• While the MLP adapter-based semantic injection achieves higher consistency and ControlNet

---

[1]https://klingai.kuaishou.com/

[2]https://runwayml.com/ai-tools/gen-2-text-to-video

[3]https://pika.art/home

demonstrates more dynamic motion, both methods fall short in certain metrics. Although Control-Net excels in dynamic motion, outperforming other methods by a significant margin, it struggles with subject consistency, particularly as observed subjectively (as shown in Appendix, Figure 10). The adaptive norm method, however, strikes a more balanced performance across all criteria.

● As depicted in Figure 6, the first sub-figure presents a clip of the AR code-reconstructed video, while the second shows the baseline without AR codes. The third to sixth sub-figures correspond to videos with AR codes injected into the last 14, first 3, 8 and 14 layers of the DiT model (with 28 layers in total) respectively. Injecting AR codes into the last 14 layers provides insufficient layout information, resulting in a video similar to the baseline. In contrast, injecting codes into the first 3, 8, and 14 layers ensures that the layout information in the generated video aligns with AR codes, with greater control achieved as more layers are involved. Similar trends can be found in Table 3, injecting AR codes into the first 14 layers produces the best dynamic degree and aesthetic quality.

● To further improve the robustness of the DiT model, two approaches are applied: adding random noise to the latent feature $F$ and employing uncertainty sampling as discussed in Section 2.3. Both methods improve the dynamic motion in generated videos, with uncertainty sampling further enhancing aesthetic and image quality.

*"A breathtaking aerial view of a rocky coastline. The coastline is a mosaic of small rocks and boulders. The deep blue water of the sea crashes against the shore, creating a frothy white foam that contrasts beautifully with the surrounding landscape."*

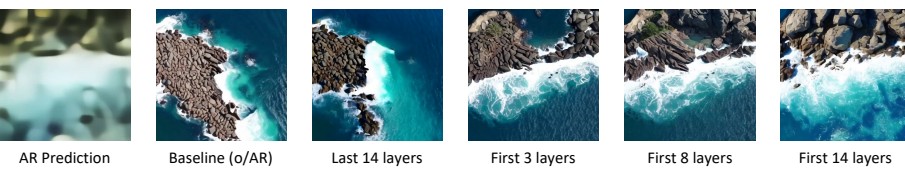

| AR Prediction | Baseline (o/AR) | Last 14 layers | First 3 layers | First 8 layers | First 14 layers |

Figure 6: Effects of incorporating layout information provided by AR codes into different layers.

Table 3: Ablation study on 1) the semantic injection module, encompassing its architecture and the number of DiT layers inserting this module; and 2) the training strategies, using coarse visual latent tokens (a latent VAE with a higher compression ratio. The compression ratio of the latent VQ-VAE for AR model is $4\times8\times8$), introducing Gaussian random noise to the latent features $F$ and performing uncertainty sampling with $F$.

| Compress Ratio in DiT | Fusion Architect | Number of Layers | Gaussian Noise | Uncertainty Sampling | Subject Consist | Background Consist | Motion Smooth | Dynamic Degree | Aesthetic Quality | Imaging Quality |
|---|---|---|---|---|---|---|---|---|---|---|
| | Baseline OpenSora-1.2 | | | | 97.79 | 97.86 | **99.36** | 19.00 | 52.46 | 61.19 |
| $4\times8\times8$ | Adaptive Norm | 14 | ✗ | ✗ | 94.71 | 96.22 | 99.07 | 42.00 | 53.91 | 58.50 |
| | ControlNet | 14 | ✗ | ✗ | 95.81 | 96.58 | 99.10 | **46.00** | 55.18 | 62.89 |
| | MLP Adapter | 14 | ✗ | ✗ | 97.99 | 97.97 | 99.31 | 21.00 | 56.40 | 64.74 |
| $4\times16\times16$ | Adaptive Norm | 3 | ✗ | ✗ | 97.56 | 97.71 | 99.27 | 28.00 | 55.32 | 65.03 |
| | Adaptive Norm | 8 | ✗ | ✗ | 97.29 | 97.55 | 99.21 | 31.00 | 55.55 | 65.09 |
| | Adaptive Norm | 14 | ✗ | ✗ | 97.40 | 97.72 | 99.24 | 32.00 | 56.78 | 65.08 |
| | Adaptive Norm | 14 | ✓ | ✗ | 97.04 | 97.35 | 99.20 | 34.00 | 54.98 | 64.21 |
| | Adaptive Norm | 14 | ✗ | ✓ | 97.39 | 97.55 | 99.24 | 34.00 | **56.90** | **65.33** |

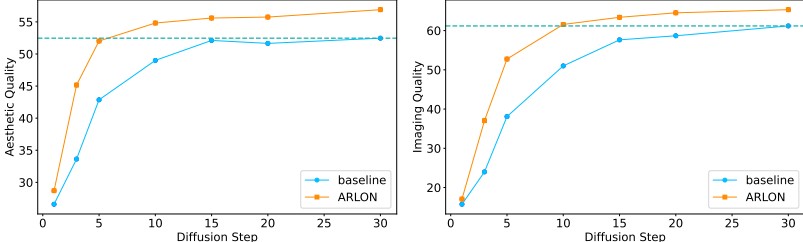

Figure 7: Aesthetic quality and imaging quality as a function of denoising steps.

## 4.4 ANALYSIS

We conduct a detailed analysis of ARLON's inference efficiency compared to OpenSora-V1.2. The AR codes, which capture key spatial information, serve as an effective initialization, significantly accelerating the DiT model's denoising process. Figure 7 illustrates the curves representing aesthetic quality and imaging quality, as a function of the number of denoising steps. Notably, ARLON

achieves the similar performance in just 5 to 10 steps, compared to the 30 steps required by the baseline. Taking the inference of a 68-frame video on a single A100 as an example, the OpenSora-V1.2 requires about 47 seconds for 30 steps of inference, while our DiT model equipped with the semantic injection module takes approximately 11 to 19 seconds for 5 to 10 steps, with additional approximately 6 seconds for the AR codes inferring. This results in a relative efficiency improvement of 47-64% over OpenSora-V1.2. Figure 8 provides a visual comparison between ARLON and the baseline OpenSora-V1.2, using 3 and 5 denoising steps. The results clearly show that ARLON generates videos with significantly higher quality.

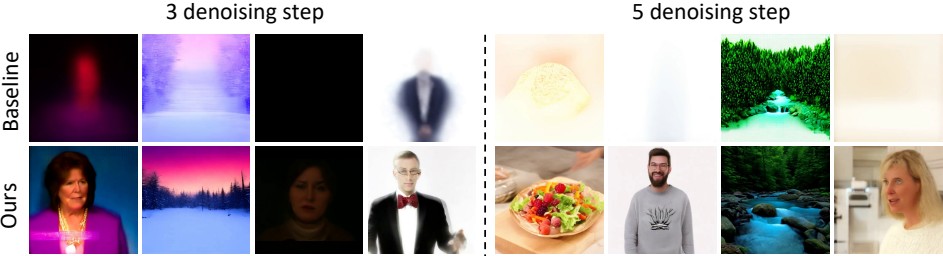

Figure 8: Generated videos of ARLON and OpenSora-V1.2 with 3 and 5 denoising steps.

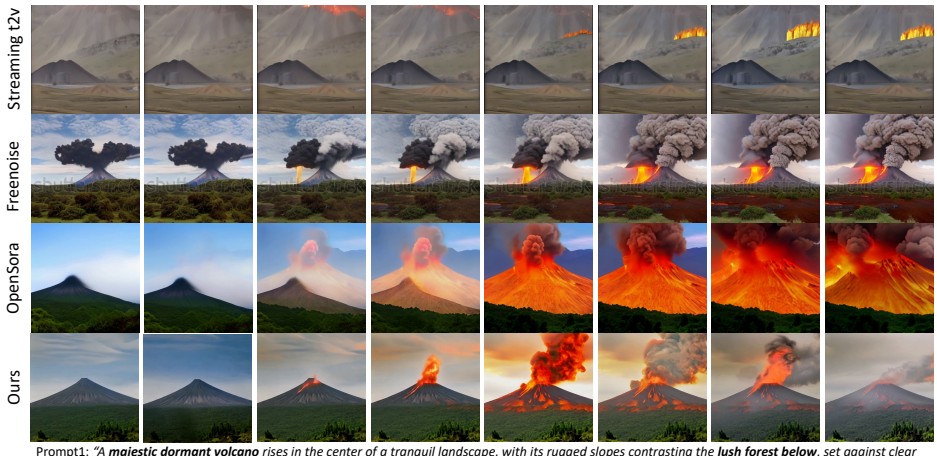

Prompt1: *"A **majestic dormant volcano** rises in the center of a tranquil landscape, with its rugged slopes contrasting the **lush forest below**, set against clear skies and rolling hills, evoking a sense of solitude and timeless serenity."* - - - - - - - - - - - - -> *(transitions to)*
Prompt2: *"An **erupting volcano** dominates the scene, with **fiery lava and ash** contrasting **the lush forest** below, as **dark clouds** and **lightning fill** the sky, capturing the chaos and power of nature's fury. "*

Figure 9: Qualitative comparisons of the videos generated using progressive prompts.

## 4.5 APPLICATIONS

One practical application of ARLON is long video generation using progressive text prompts. The procedure is as follows: assuming two text prompts, $X_1$ and $X_2$, are given, the corresponding AR codes $Q_1$ are first generated based on $X_1$. Then, the last frame of $Q_1$ is used as a condition, along with $X_2$, to generate the corresponding AR code $Q_2$. Finally, the DiT model utilizes $(X_1, Q_1)$ and $(X_2, Q_2)$ to generate the video. A qualitative result is presented in Figure 9. Long videos generated by other models either remain unchanged after a prompt transition or change drastically. In contrast, our model transitions seamlessly, maintaining consistency throughout the entire video.

## 5 CONCLUSION

In this paper, we propose ARLON, a novel framework that boosts diffusion Transformers with autoregressive (**AR**) models for long (**LON**) video generation. Utilizing a latent VQ-VAE, the input latent of the DiT model is compacted and highly quantized into discrete tokens for the training and inference of the AR model. To integrate coarse spatial and temporal features into the DiT model, an adaptive norm based semantic injection module is proposed. To improve the robustness of ARLON, the DiT model is trained with coarser visual latent tokens incorporated with an uncertainty sampling module. Massive experiments demonstrate that our ARLON model delivers state-of-the-art performance on text-based long video generation. Detailed analysis of the inference efficiency and qualitative results of progressive prompt based long video generation are also provided.

ACKNOWLEDGEMENT

Zongyi Li was supported by the Natural Science Foundation of China (62372203, 62302186) and the Major Scientific and Technological Project of Shenzhen (202316021).

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

# A APPENDIX

## A.1 PRELIMINARIES

In this section, we introduce the preliminaries of a stable diffusion model, which operates the diffusion process in a latent space for computationally efficient. In this model, the image $x$ is mapped into a compressed latent space $z = \mathbb{E}(x)$ via a pre-trained auto-encoder, such as VQGAN or VQVAE. In the forward process, random noise is gradually added to the latent space, formulated as:

$$q\left(z_t \mid z_{t-1}\right) = \mathcal{N}\left(z_t; \sqrt{\alpha_t}z_{t-1}, (1 - \alpha_t)\mathbf{I}\right), \tag{8}$$

where $t \in \{1, \dots, T\}$, $T$ is the number of time steps during the diffusion process. and $q\left(z_t \mid z_{t-1}\right)$ is the noised $z_t$ at $t$ step given $z_{t-1}$, and $(1 - \alpha_t)$ denotes the noise strength. Alternatively, we can formulated $z_t$ from $z_0$ as follows:

$$z_t = \sqrt{\bar{\alpha}_t}z_0 + \sqrt{1 - \bar{\alpha}_t}\epsilon_t, \epsilon_t \sim \mathcal{N}(0, \mathbf{I}), \tag{9}$$

where $\bar{\alpha}_t = \prod_{i=1}^{t} \alpha_i$. The denoising process involves learning a reverse diffusion process to iteratively remove the noise added during the forward process. This is achieved by training a neural network to predict the original latent representation from the noisy version by minimizing:

$$l_\epsilon = \|\epsilon - \epsilon_\theta\left(z_t, t, c\right)\|_2^2, \tag{10}$$

where $c$ denotes the conditional textual description, for the inference, random noise is sampled from Gaussian distribution, and DDIM is utilized for denoising a latent representation, followed by a VAE decoder to reconstruct the image.

## A.2 EXTENDED MODEL COMPARISONS

Table 4: Comparative Analysis of Model Parameters, Training Memory, Inference Speed, and Computational Efficiency for OpenSora-V1.2 and ARLON. The batch size for training both the AR and DiT models is set to 2x68-frame 512x512 video clips, and the inference is evaluated by generating a 68-frame (68f) or 578-frame (578f) 512x512 video. Both the training and inference processes are executed on a single 40G A100 GPU.

| Method | OpenSora-V1.2 (baseline) | ARLON (ours) |
|---|---|---|
| Param. (AR) | - | 192M |
| Param. (DiT) | 1.2B | 92M (trainable) + 1.2B (frozen) |
| Param. (3D VAE) | 384M | 384M |
| Param. (latent VQ-VAE) | - | 30M |
| Training Memory | 36007M | 7701M (AR); 36815M (DIT) |
| Inference Memory | 24063M | 2269M (AR); 25215M (DIT) |
| Inference speed (68f) | 47.3 s | 5.7s (AR)+18.9s (DiT) |
| Inference speed (578f) | $47.3 \times 11$ s | 57.2s (AR) + (18.9 $\times$ 11) s (DiT) |
| Inference FLOPs (68f) | 42626G $\times$ 30 (step) | 200G (AR) + 46461G$\times$10 (step) (DiT) |
| Inference FLOPs (578f) | 42626G $\times$ 30 (step) $\times$ 11 (times) | 1547G (AR) + 46461G$\times$10 (step) $\times$ 11 (times) (DiT) |

We have conducted a comparative analysis of model parameters, training memory, inference speed, and computational efficiency for OpenSora-V1.2 and ARLON, as shown in Table 4. From the results in the table, we can observe that:

• The increase in the number of parameters (192M + 92M + 30M) of ARLON is minimal compared to the 1.2B parameters of the baseline, OpenSora-V1.2. This is primarily due to our adoption of an efficiency adapter approach during training, which enables us to introduce fewer parameters while preserving performance. Additionally, the latent VQ-VAE operates within a compressed latent space, and the AR model is training and generates highly quantized tokens, both of which significantly contribute to the minimized parameter requirements.

• It is evident that our method does not require significant additional memory and computational resources compared to the baseline model. Specifically, during the training and inference phases, there are 2.2% and 4.8% relative increases respectively (the AR and DiT models can be trained independently).

• Conversely, by leveraging the AR code as an efficient initialization, our model is capable of generating high-quality videos with significantly fewer steps. Consequently, our inference time and

total FLOPs are superior to those of the baseline, thereby significantly accelerating the denoising process. Specifically, our model achieves a 48-49% relative improvement in inference speed and a 64% relative reduction in computational FLOPs for 68-frame or 578-frame video generation (578 can be expressed as $68 \times 11 - 17 \times 10$. Here, 11 represents the number of times the DiT model generates 68-frame video segments, while 17 signifies the number of frames in the conditioned video. Additionally, 10 indicates the number of times the DiT model generates videos under specific conditions).

## A.3 EFFECT OF TRAINING DATA SIZE FOR DiT MODEL

Table 5: Comparative Analysis of Different Training Data Sizes.

| Models | Subject Consist | Background Consist | Motion Smooth | Dynamic Degree | Aesthetic Quality | Imaging Quality |
|---|---|---|---|---|---|---|
| Openvid-1M | 95.02 | 96.35 | 98.16 | 30.00 | 52.34 | 59.15 |
| Openvid-HQ 0.4M | 97.78 | 97.83 | 99.25 | 30.00 | 55.42 | 64.11 |
| Openvid-HQ 0.4M+Mixkit 0.3M | 97.39 | 97.55 | 99.24 | 34.00 | 56.90 | 65.33 |

In Table 5, we further illustrate the impact of varying training data sizes. Firstly, when comparing our model's performance on the OpenVid 1M dataset with that on OpenVid-HQ (which contains higher quality videos, totaling 0.4M), we observed a marked improvement in our model's performance on OpenVid-HQ. This indicates that the quality of the data plays a crucial role in the task of video generation. Furthermore, when we combined the OpenVid-HQ and Mixkit (the quality of videos is also high) datasets as our training set (approximately 0.7M), improvements in both quality and dynamic degree are obtained. This suggests that in the context of video generation, prioritizing high-quality videos while also utilizing a larger dataset can effectively enhance the overall quality of generated videos.

## A.4 LIMITATIONS

Although ARLON achieves state-of-the-art performance in long video generation, it also exhibits some specific constraints. First, ARLON is built upon OpenSora-V1.2, which potentially caps the upper limit of video quality. Nonetheless, this limitation can be mitigated by substituting the DiT model with more advanced alternatives, such as CogVideoX-5B or MovieGen. Second, if we aim to train ARLON at 2K resolution, the sequence length of AR codes will become excessively long, making both training and inference impractical. Viable solutions involve employing a higher compression ratio in VQ-VAE, or selectively retaining essential information while disregarding irrelevant details. Additionally, for the AR model, parallel prediction emerges as an alternative approach.

We also have presented some failure cases, as shown in Figure 16. For example, generating hand movements such as applying makeup or eating is challenging for creating realistic videos that conform to the physical world, especially regarding hand details. Our future research endeavors will delve into addressing these issues.

## A.5 BROADER IMPACT

Synthetic video generation is a powerful technology that can be misused to create fake videos or videos containing harmful and troublesome content, hence it is important to limit and safely deploy these models. From a safety perspective, we emphasize that the training data of ARLON are all open-sourced, and we do not add any new restrictions nor relax any existing ones to OpenSora-V1.2. If you suspect that ARLON is being used in a manner that is abusive or illegal or infringes on your rights or the rights of other people, you can report it to us.

## A.6   EXTRA EXAMPLES

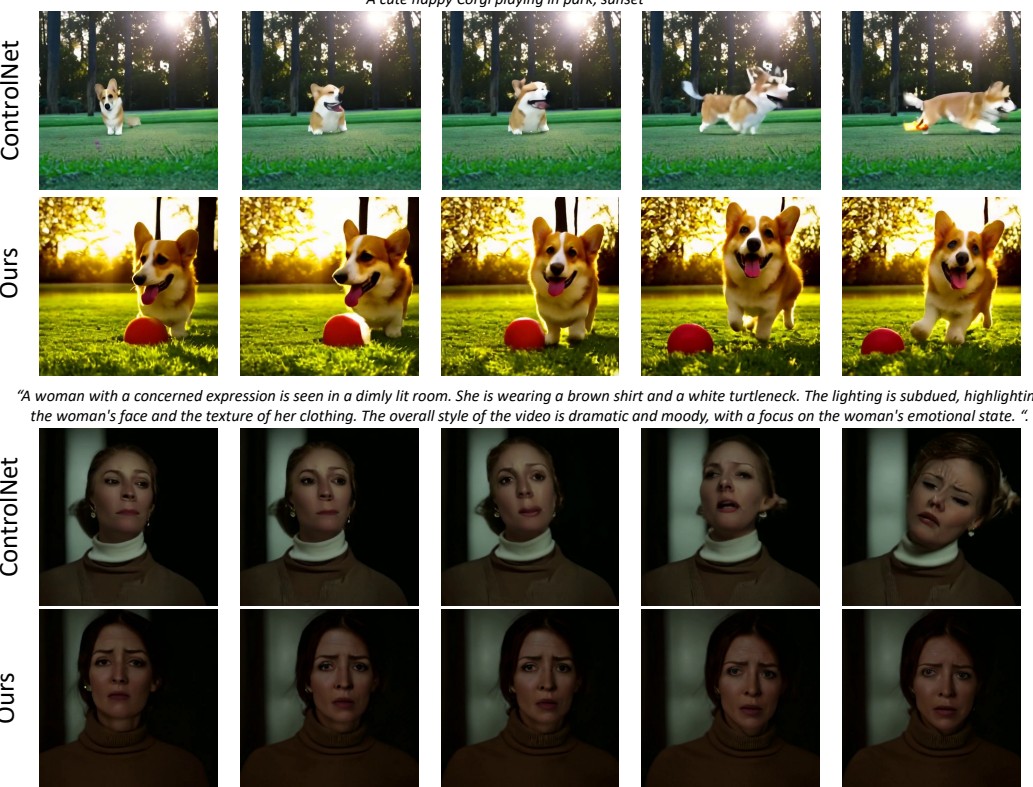

Figure 10: Ablation study on different semantic injection approaches. Although ControlNet has higher dynamism, it also produces more distortions, such as the severe deformation of the dog in the left video and the facial distortions in the right video.

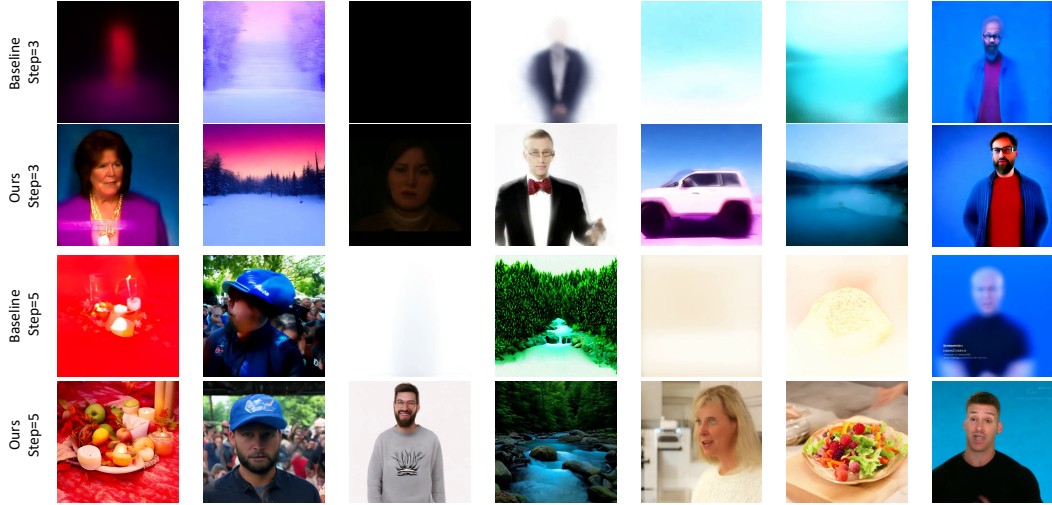

Figure 11: More generated videos of ARLON and OpenSora-V1.2 with 3 and 5 denoising steps, as a complement to Figure 8.

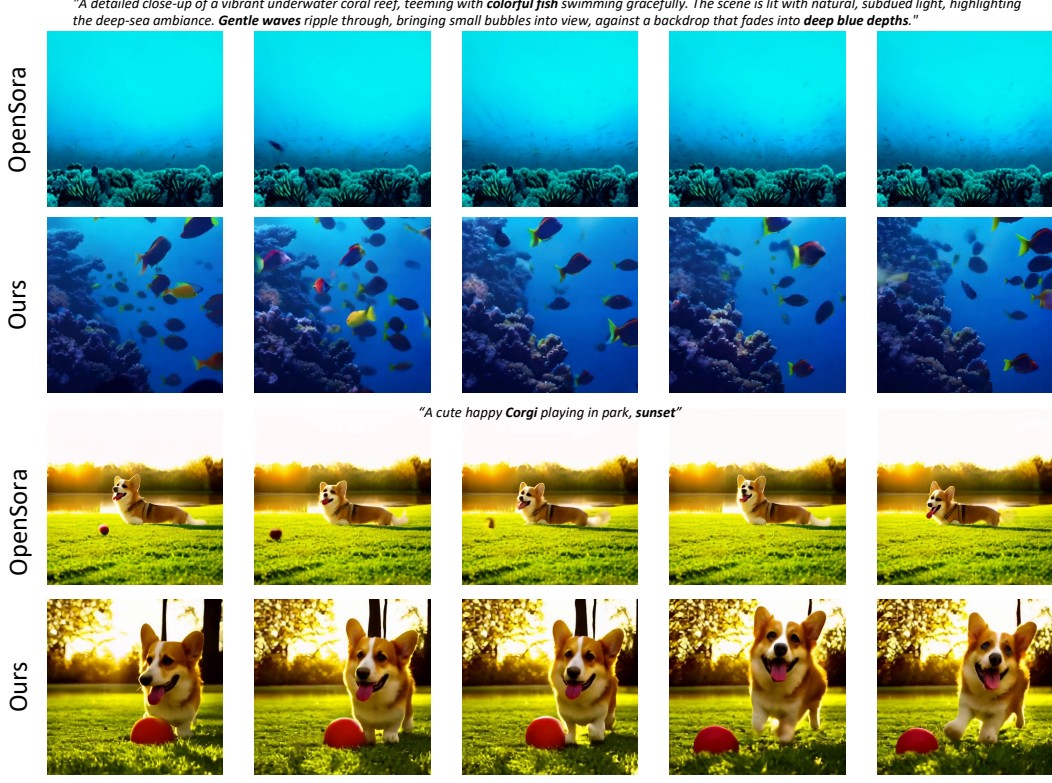

Figure 12: Examples of text to videos generation of ARLON and OpenSora-V1.2.

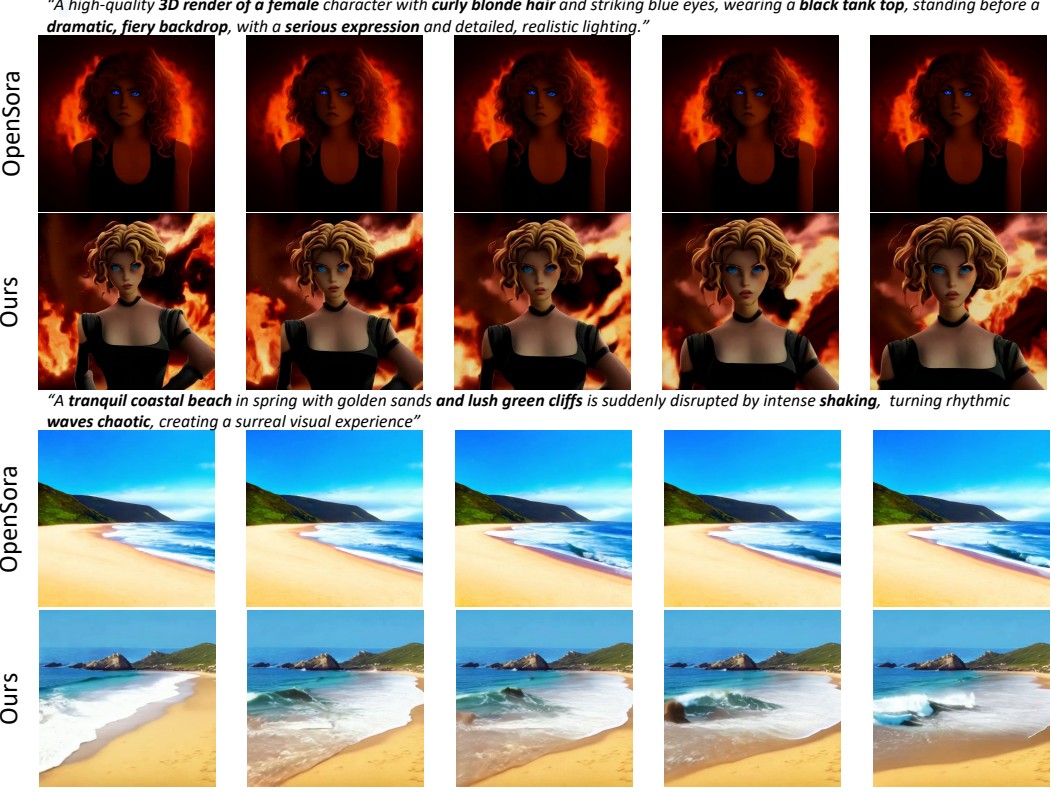

Figure 13: More text to videos generation of ARLON and OpenSora-V1.2.

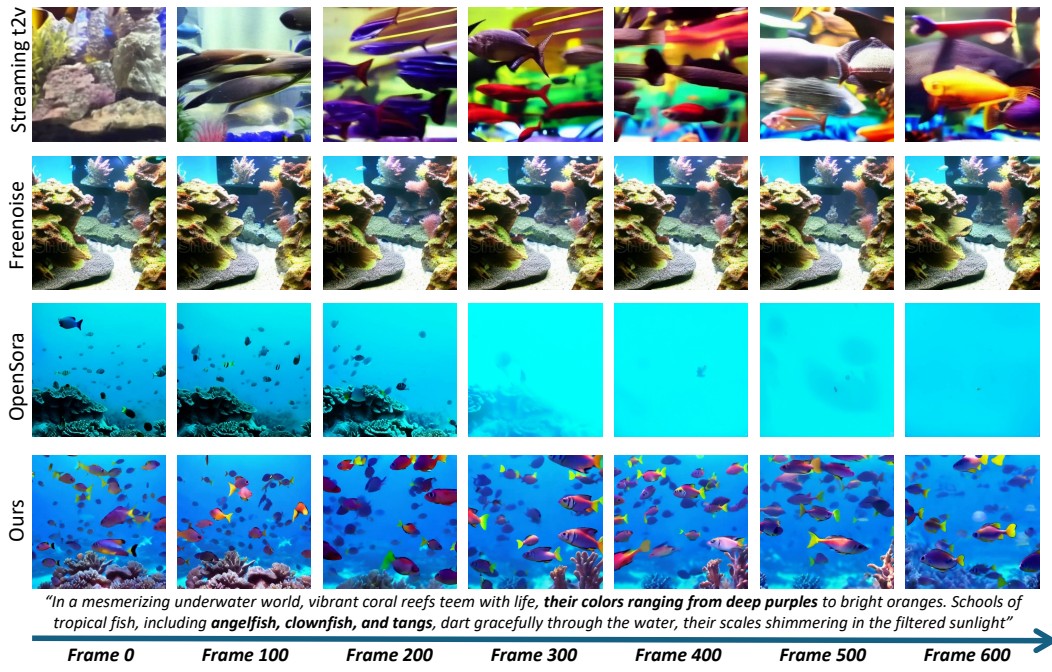

Figure 14: Examples of text to long video generation (600 frames) using FreeNoise, StreamingT2V, OpenSora-V1.2, and ARLON.

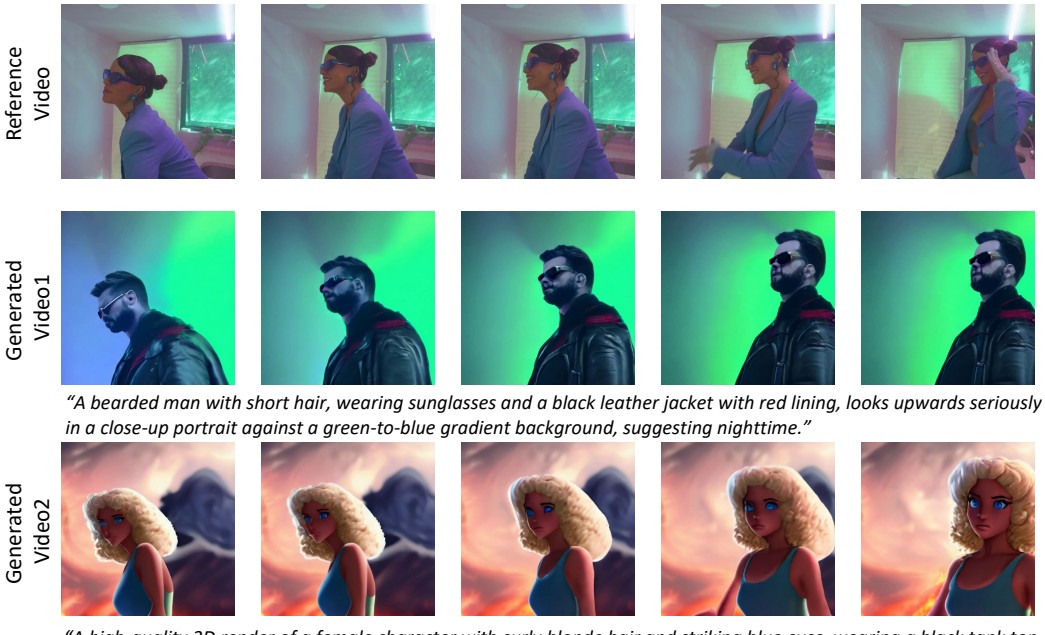

Figure 15: Examples of Video Condition Generation: We extract the coarse latent from reference videos and replace it with our semantic latent, which is then injected into the diffusion process. The generated video maintains the same pose and position as the reference video while ensuring consistency with the provided textural prompts.

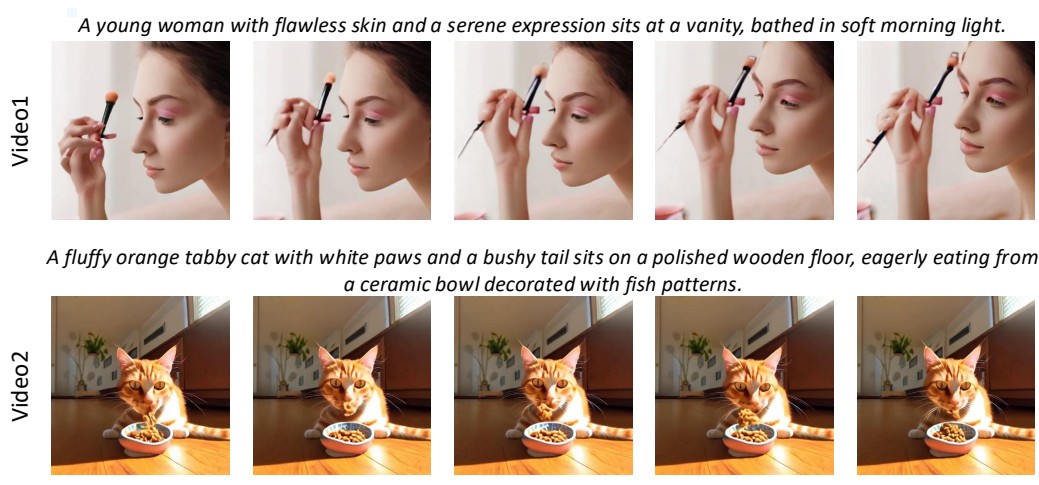

Figure 16: Examples of failure cases.

