# OpenReview forum: "ARLON: Boosting Diffusion Transformers with Autoregressive Models for Long Video Generation"
_ICLR.cc/2025/Conference — ICLR 2025 Poster_

### Official Review · Reviewer_45zF · 2024-11-02

**Soundness:** 3
**Presentation:** 3
**Contribution:** 3
**Rating:** 6
**Confidence:** 4

**Summary:**

This paper proposes a new text-to-video(T2V) framework consisting of autoregressive (AR) Transformers and Diffusion Transformers (DiT). Based on the input text prompt, the AR model predicts quantized visual tokens of a latent VQ-VAE nested within the 3D VAE of the DiT model. The coarse latent, reconstructed from the predicted tokens, serves as semantic condition to guide the DiT through adaptive normalization for video generation. To mitigate the effect of error introduced from AR inference, the authors introduce two noise-resilient strategies during DiT training, using coarser latent tokens and uncertainty sampling to make the semantic condition noisier.

**Strengths:**

1.This paper innovatively combines the strengths of autoregressive (AR) Transformers and Diffusion Transformers (DiT) for generating long video with rich dynamic motion.
2. To mitigate the effect of error introduced from AR inference to DiT, the authors introduce two noise-resilient strategies during DiT training.
3. The paper is written and presented clearly and easy to follow.
4. The long video results of the proposed method show improvement on dynamic degree, and long video generation results using progressive text prompts are more consistent throughout the entire video.

**Weaknesses:**

1. From Table 1, the proposed method lags behind compared methods in many metrics other than dynamic degree, such as Imaging Quality and Subject Consistency. In Table 2, the dynamic degree of proposed method (50.42) is significantly lower than that of the StreamingT2V(85.64).
2. From the demo videos on the webpage, there is some room for improvement for the proposed method compared to others. For example, the result of "A teddy bear is swimming in the ocean." lacks of subject consistency, and its motion is not realistic, which may be consistent with the quantitative results in Table 1 and Table 2.
3. Although the authors introduce noise-resilient strategies for the DiT model training to mitigate the error issue from AR inference, I am concerned that these strategies cannot truly simulate the error of AR inference, which may limit the model performance.

**Questions:**

1. In Figure1, the previously generated video seems to be used as a reference for the subsequent generation, but this is not illustrated in Figure 2. Does the DiT generate videos in an autoregressive way?
2. In Table 1 and Table 2, do the higher scores of metrics indicate better performance?
3. For long video generation in section 4.2 as well as the demo videos on the webpage, the authors only compare their proposed method with open-source text-to-long video generation models. Why not compare with the commercial closed-source text-to-video generation models like in Table 1?

---

> ### Author Response · Authors · 2024-11-22
> **Response to Reviewer 45zF - Part 1**
>
> We would like to express our gratitude to the reviewer for highlighting the innovative combination of AR and DiT for long video generation, the noise-resilient strategies during DiT training, the clear presentation of the paper, and the improvements in dynamic degree and consistency of long video generation with progressive text prompts. We would also like to thank the reviewer for the constructive suggestions and comments which will be responded to one-by-one below.
>
>
>
> **W1.** From Table 1, the proposed method lags behind compared methods in many metrics other than dynamic degree, such as Imaging Quality and Subject Consistency. In Table 2, the dynamic degree of the proposed method (50.42) is significantly lower than that of the StreamingT2V(85.64).
>
> **Response:** We thank the reviewer for the insightful comments!
>
> Table 2 (Table 1 in the original manuscript) presents the metrics for short video generation. Due to the presence of many closed-source algorithms and the inconsistency in model sizes and training datasets, it is challenging for any single algorithm to excel across all metrics. Our approach is based on OpenSora, and it significantly **outperforms the baseline OpenSora-V1.2 on eight out of eleven metrics** selected from VBench, with notable improvements in dynamic range and aesthetic quality. Moreover, it delivers competitive performance on the remaining three metrics, while also **accelerating the video generation process**. Notably, our method focuses on **long video generation**, where it excels in maintaining both consistency and dynamic range over extended sequences.
>
> Regarding long video generation of Table 1, (Table 2 in the original manuscript), while StreamingT2V achieves a high dynamic degree, this is primarily due to its chaotic scene transitions and frequent object movements, which often lead to instability. As shown in Figures 4 (Figure 5 in the original manuscript) and 14, StreamingT2V demonstrates substantial changes in object motion and background over time, with sudden object switches, abrupt scene changes, and screen disruptions. Consequently, it exhibits a high degree of dynamism but **suffers from low consistency in both subject and background**. In contrast, our approach ensures consistent backgrounds and subjects while achieving high dynamic scores in long video generation, **maintaining both coherence and dynamism over extended video sequences**.
>
>
>
> **W2.** From the demo videos on the webpage, there is some room for improvement for the proposed method compared to others. For example, the result of "A teddy bear is swimming in the ocean." lacks of subject consistency, and its motion is not realistic, which may be consistent with the quantitative results in Table 1 and Table 2.
>
> **Response:** We thank the reviewer for the insightful comment!
>
> We would like to clarify that the demo of "A teddy bear is swimming in the ocean." is for long video generation. This aligns with the quantitative results presented in Table 1 (Table 2 in the original manuscript), where our model, ARLON, demonstrates the **highest levels of subject consistency and background consistency, along with superior dynamics**, when compared to other open-source models.
>
> However, we acknowledge that there is indeed potential for enhancement in our models.  We are confident that integrating a more robust and advanced DiT model could yield even superior results. Moving forward, we intend to persistently update and investigate improvements to our methodology.

---

> ### Author Response · Authors · 2024-11-22
> **Response to Reviewer 45zF - Part 2**
>
> **W3.** Although the authors introduce noise-resilient strategies for the DiT model training to mitigate the error issue from AR inference, I am concerned that these strategies cannot truly simulate the error of AR inference, which may limit the model performance.
>
> **Response**: We thank the reviewer for the insightful comment!
>
> We would like to clarify that
>
> 1. There are two noise-resilient strategies are utilized in our work, coarser visual latent tokens and uncertainty sampling. We acknowledge that the approach of coarser visual latent tokens does not fully simulate the errors inherent in AR inference, as it is a holistic simulation, whereas the errors introduced by AR inference occur at the token level. However, the **uncertainty sampling** method introduces noise at the token level, which is a simulation of the errors introduced by AR inference to some extent.
> 2. In our preliminary experiments, we employed a more direct simulation strategy, which involved randomly replacing 30-50% of the AR codes with incorrect ones during training. However, this approach yielded unsatisfactory results. We attribute this to the excessive errors introduced by such a simulation method, which confounded the DiT model's ability to discern the underlying relationship between the AR codes and the corresponding videos. **In comparison, uncertainty sampling serves as a notably more effective approach.**
> 3. In fact, **we do not necessarily need to fully simulate the errors introduced during AR inference**. Instead, we aim to enable the DiT model to follow the information provided by AR while simultaneously being tolerant of the errors introduced during the AR inference phase, as we discussed in Section 2.3 ("*To tolerate the errors inevitably introduced during AR inference, we implement two noise-resilient training strategies: coarser visual latent tokens and uncertainty sampling.*"). Specifically, for utilizing the coarser visual latent tokens, the coarser the AR code, the less information it contains, and the more the DiT model needs to generate. Even if the AR model provides an incorrect code, the DiT model does not blindly follow it but rather attempts to generate videos that align with ground truth videos. On the other hand, introducing noise through the uncertainty sampling approach introduces noise directly at the token level, offering a more straightforward method to improve the DiT model's robustness against errors.
>
> In addition, we have revised the content of Section 4.3 to avoid any confusion for the readers ("which could simulate the errors" -> "which could make the DiT model tolerate the errors"). The specific changes made are highlighted in red. We once again express our gratitude to the reviewer for the constructive questions, which have helped us improve the quality of the paper.
>
>
>
> **Q1.** In Figure 1, the previously generated video seems to be used as a reference for the subsequent generation, but this is not illustrated in Figure 2. Does the DiT generate videos in an autoregressive way?
>
> **Response:** We thank the reviewer for the question!
>
> Yes, the DiT model in ARLON generates videos in an autoregressive approach. The reason why it is not illustrated in Figure 2, is that Figure 2 is the overview of the training stage of ARLON, while the autoregressive way is used in training.
>
> As described in Section 2.3 "Training Strategy", *to enable this autoregressive approach, we randomly unmask certain frames, keeping them noise-free to serve as conditioning frames.* This allows the diffusion model to consider preceding frames as conditions during inference.
>
>
>
> **Q2.** In Table 1 and Table 2, do the higher scores of metrics indicate better performance?
>
> **Response:** We thank the reviewer for the question!
>
> Yes, in Table 1 and Table 2, the higher scores of metrics indicate better performance.
>
> To improve the readability of Table 1 and Table 2, we have added the sentence "the higher scores of metrics indicate better performance." to the table captions.
>
>
>
> **Q3.** For long video generation in section 4.2 as well as the demo videos on the webpage, the authors only compare their proposed method with open-source text-to-long video generation models. Why not compare with the commercial closed-source text-to-video generation models like in Table 1?
>
> **Response:** We thank the reviewer for the question!
>
> Commercial closed-source text-to-video models, like Kling, typically produce short clips limited to around 4-10 seconds, lacking continuous scenes over 30 seconds. They often rely on multiple prompts, resulting in stitched rather than seamless videos. Therefore, we focus on comparing open-source text-to-long video generation models.

---

> ### Author Response · Authors · 2024-11-26
>
> Dear Reviewer 45zF,
>
> We sincerely appreciate your thoughtful feedback, which we have carefully addressed in our response. We hope our responses effectively address your questions. If you have any further inquiries or suggestions, we would be delighted to engage in further discussion before the conclusion of the discussion period. Your insights have been invaluable in enhancing our work, and we look forward to your thoughts on the revised version.
>
> Best regards

---

> ### Comment · Reviewer_45zF · 2024-11-28
>
> Thanks for the authors' reply, my raised concerns have been mostly addressed. After I further read the author's responses to the questions I raised, as well as the communication between the author and other reviewers, I am more confident in the technical novelty and value of this paper. I tend to slightly raise my score.

---

> > ### Author Response · Authors · 2024-11-29
> >
> > Thank you very much for your thoughtful feedback and for acknowledging that our responses have addressed your concerns.  We are truly grateful for your acknowledgment of the technical novelty and value of our paper, as well as your inclination to raise the score.
> >
> > If you feel that an adjustment to your score or level of confidence is appropriate, we would greatly appreciate it. Additionally, please feel free to reach out if you have any further questions or comments; your insights are invaluable to us.
> >
> > Thank you once again for your time and consideration.
> >
> > Best regards,
> >
> > Authors

---

### Official Review · Reviewer_gQNT · 2024-11-04

**Soundness:** 2
**Presentation:** 3
**Contribution:** 3
**Rating:** 8
**Confidence:** 3

**Summary:**

This paper proposes to leverage autoregressive models to guide the training of diffusion transformers for text-to-video generation task. The proposed framework incorporate with a latent VQ-VAE, coarser visual latent tokens and a uncertainty sampling module to connect DiT and AR models and inject the information from AR models to DiT training. Massive experiments are conducted with abundant quantitative metrics and visualizations are reported to demonstrate the performance of the proposed model.

**Strengths:**

- The idea of make DiT and AR models working in one latent space is novel, and many technical improvements are designed to bridge their gap.

- The proposed method reaches a large reduction of denoising step of comparable generation quality.

**Weaknesses:**

- It's better to use bold fonts and underscores in Table 1. According to the listed numbers, the proposed method doesn't reach top 3 in many columns. And the reproduced notation is missing.

**Questions:**

- What is the motivation of using a latent VQ-VAE nested inside a pretrained 3D Autoencoder, instead of training a single-stage pixel-to-latent tokenizer? And what is the additional computational cost or gain in comparison?

- The proposed coarse latent token with different compression ratio, while how is this ablated in Table 3 or any other ablation studies? (Does the 4×8×8 row refers to both the same scale training and the 4×16×16 rows refer to different scale training?)

---

> ### Author Response · Authors · 2024-11-22
> **Response to Reviewer gQNT - Part 1**
>
> We would like to express our gratitude to the reviewer for highlighting the novelty and the acceleration capacity of our ARLON. We would also like to thank the reviewer for the constructive suggestions and comments which will be responded to one-by-one below.
>
>
>
> **W1.** It's better to use bold fonts and underscores in Table 1. According to the listed numbers, the proposed method doesn't reach the top 3 in many columns. The reproduced notation is missing.
>
> **Response:** We thank the reviewer for the insightful comments!
>
> We will use bold fonts and underscores for the best numbers in Table 2 (Table 1 in the original manuscript). All the compared results in Table 2 (Table 1 in the original manuscript) are numbers reported in their respective papers, and we will remove the sentence 'The reproduced notation is missing.'. We also would like to clarify the results compared with other methods as follows:
>
> 1. The performance of ARLON is also outstanding in the field of short video generation. Our ARLON is built upon OpenSora-V1.2, making it a fair comparison to OpenSora-V1.2. As evidenced by Table 2 (Table 1 in the original manuscript), it is observed that ARLON significantly outperforms the baseline OpenSora-V1.2 on eight out of eleven metrics selected from VBench, with notable improvements in dynamic range and aesthetic quality. Moreover, it delivers competitive performance on the remaining three metrics, while also accelerating the video generation process.
> 2. Due to the presence of many closed-source algorithms and the inconsistency in sizes of model parameters and training datasets, it is challenging for any single model to excel across all metrics.
> 3. ARLON focuses on long video generation, where it excels in maintaining both consistency and dynamic range over extended sequences as highlighted in Table 1 (Table 2 in the original manuscript). In contrast, Table 2 (Table 1 in the original manuscript) provides the metrics for short video generation.
> 4. To better present results and emphasize our strengths, we have divided Table 2 (Table 1 in the original manuscript) into two sections. The lower section presents the results of ARLON and OpenSora-V1.2, with our superior results highlighted with specific improvements. The upper section, meanwhile, displays the results of other text-to-video models.

---

> ### Author Response · Authors · 2024-11-22
> **Response to Reviewer gQNT - Part 2**
>
> **Q1.** What is the motivation of using a latent VQ-VAE nested inside a pretrained 3D Autoencoder, instead of training a single-stage pixel-to-latent tokenizer? And what is the additional computational cost or gain in comparison?
>
> **Response**: Thank you for the insightful question.
>
> The motivation for using a latent VQ-VAE nested inside a pretrained 3D Autoencoder, rather than training a single-stage pixel-to-latent tokenizer, is based on several key considerations:
>
> 1. *Leveraging the Benefits of Pretrained Models*: Initiating with a large-scale data pretrained model is an efficient and effective strategy. OpenSora-v1.2, a text-to-video DiT-based model based on a 3D VAE, has been trained on an extensive corpus of video data. It stands out as one of the most effective open-source models, boasting a significant following and serving as our baseline model. Following OpenSora-v1.2's setup, we leverage its pretrained 3D VAE.
> 2. *Ensuring a Consistent Latent Space between AR and Diffusion Models*:  Constrained with the latent space of the pre-trained 3D VAE and the OpenSora-v1.2, we need to convert the sequence of features generated by the 3D VAE encoder to discrete token sequence for AR model training, as well as a reverse conversion for inference. Aiming this, we introduce a nested VQ-VAE to align the semantic space of the AR and diffusion models.
> 3. *Balancing Information Density and Learning Complexity*: The VQ-VAE, when configured with an appropriate compression ratio, efficiently condenses the input latent space into a compact and highly quantized set of visual tokens, while retaining the essential information. This allows the AR model to focus on predicting coarse information rather than grappling with fine-grained details, thereby enhancing learning efficiency.
>
> In addition, we would like to clarify the additional computation cost and gains:
>
> Although the introduced VQ-VAE may incur some additional computational overhead, we believe this cost can be justified.
>
> 1. The quantization process of VQ-VAE substantially reduces the amount of information that the subsequent AR model needs to process, thereby alleviating the overall computational burden. Furthermore, the pretrained 3D VAE can accelerate the training process and decrease the number of iterations required, ultimately conserving computational resources in the long term. Consequently, despite **a small potential short-term rise** in computational cost, we are confident that this design choice is **advantageous in terms of overall efficiency and performance.**
> 2. Furthermore,  since the latent VQ-VAE operates within the latent space, its parameters are significantly smaller than the 3D VAE.
>
> | **Method**             | **OpenSora-V1.2 (baseline)** | **ARLON (ours)** |
> | ---------------------- | ---------------------------- | ---------------- |
> | Param. (3D VAE)        | 384M                         | 384M             |
> | Param. (latent VQ-VAE) | -                            | 30M              |
>
> We have incorporated these results into the Appendix, which can be found in A.2. We greatly appreciate your valuable feedback and suggestions.
>
>
>
> **Q2.** The proposed coarse latent token with different compression ratio, while how is this ablated in Table 3 or any other ablation studies? (Does the 4×8×8 row refer to both the same scale training and the 4×16×16 rows refer to different scale training?)
>
> **Response:** We thank the reviewer for the insightful question!
>
> We are sorry that the column name may cause some confusion. The "4×8×8" row indicates that both the AR model and the semantic injection module employ a latent VQ-VAE with a 4×8×8 compression ratio during training. In contrast, "4×16×16" indicates that while the AR model uses the 4×8×8 compression ratio latent VQ-VAE, the semantic injection module is trained with a 4×16×16 scale. In this configuration, the DiT module is provided with a coarser latent representation, which makes the DiT model tolerate the errors introduced in the AR inference, thereby improving its robustness, and maintaining the consistency and qualities of the generated videos.
>
> To further improve the readability of Table 3, we have changed the column name "Compress Ratio" to "Compress Ratio **in DiT**", and added the description "The compression ratio of the latent VQ-VAE for AR model is 4x8x8" into the table caption.

---

> > ### Comment · Reviewer_gQNT · 2024-11-24
> >
> > Thank the authors for the detailed response. All my previous concerns have been addressed, while I want to have some more discussions on the proposed framework, especially about Figures 1 and 2:
> >
> >  - In Fig. 1, two DiT models are employed, and the differences of their roles are not well addressed. Do the numbers of frames to their left indicate there is a temporal coarse-to-fine interpolation process? In Fig. 2 (and Sec. 2.2) there is only one DiT presented.
> >
> >  - Fig. 1 needs to be overall improved. The arrows from the AR model to different DiT models should be distinguished by colors or texts indicating how they're different. The "reference" connection between the two DiT models is too concise and unclear as no other paragraphs mentions the same word.
> >
> >  - Fig. 2 also needs to be overall improved. Currently each stage or module is not clearly separate in zones. For example, the latent VQ-VAE and AR model should be in a dedicated area or full row, and so as the outer 3D VAE (middle row) and the DiT models (bottom row). The latent adapter to the right is not clear where and how it is applied and shows too many details. The blurry video frames are a bit confusing: it is mentioned that coarser latent is used for more global information, but why the output of the outer 3D VAE is still blurry, and what is it calculated w.r.t. as the ground truth?
> >
> > Thank the authors again for further information.

---

> > > ### Author Response · Authors · 2024-11-24
> > >
> > > We would like to thank the reviewer for the constructive suggestions and comments which will be responded to one-by-one below.
> > >
> > > **Q1 and Q2:** In Fig. 1, two DiT models are employed, and the differences of their roles are not well addressed. Do the numbers of frames to their left indicate there is a temporal coarse-to-fine interpolation process? In Fig. 2 (and Sec. 2.2) there is only one DiT presented. & Fig. 1 needs to be overall improved. The arrows from the AR model to different DiT models should be distinguished by colors or texts indicating how they're different. The "reference" connection between the two DiT models is too concise and unclear as no other paragraphs mentions the same word.
> > >
> > > **Response:**  We thank the reviewer for the question!
> > >
> > > As stated in the second paragraph of the Introduction, "*autoregressive approaches for long video generation with DiT models, generating successive video segments conditioned on the last frames of the previous segment.*", We also adopted this autoregressive approach, which means that **the DiT model depicted in the middle part of Figure 1 and the one in the right part are identical**.
> > >
> > > The entire inference process is as follows: 1) The AR model first generates long-term, coarse-grained discrete visual units (AR codes) in an autoregressive manner; 2) These discrete AR codes are then segmented and sequentially fed into the DiT model by the proposed semantic injection module, which autoregressively generates high-quality video segments. Specifically, the first N seconds of AR codes guide the DiT model to generate the first video segment as illustrated in the middle part of Figure 1. **The second N second of AR codes, along with the last M seconds of the first video segment, serve as the condition to generate the subsequent video segment. This process continues until the entire long video is generated.**
> > >
> > > In the training stage, as described in Section 2.3 "Training Strategy", *to enable this autoregressive approach, we randomly unmask certain frames, keeping them noise-free to serve as conditioning frames.* This allows the diffusion model to consider preceding frames as conditions during inference.
> > >
> > > We have integrated this detailed inference process into the Introduction and highlighted it in blue. In addition, the arrows connecting the AR model to the DiT model have been labeled with "first" or "second" video segment to clarify their roles, The "reference" connection has been revised as "condition" aligning with the content of the Introduction.
> > >
> > >
> > >
> > >
> > >
> > > **Q3:** Fig. 2 also needs to be overall improved. Currently each stage or module is not clearly separate in zones. For example, the latent VQ-VAE and AR model should be in a dedicated area or full row, and so as the outer 3D VAE (middle row) and the DiT models (bottom row). The latent adapter to the right is not clear where and how it is applied and shows too many details. The blurry video frames are a bit confusing: it is mentioned that coarser latent is used for more global information, but why the output of the outer 3D VAE is still blurry, and what is it calculated w.r.t. as the ground truth?
> > >
> > > **Response:**
> > > We sincerely appreciate the reviewer’s valuable feedback and have implemented the following improvements:
> > >
> > > 1. **Module Separation:** We have restructured Figure 2 into distinct sections, each dedicated to a specific component: a) the Latent VQ-VAE Compression module; b) the Autoregressive Modeling module; and c) the Semantic-Aware Condition Generation module (DIT). This layout is designed to enhance the clarity and distinguishability of the various modules.
> > > 2. **Latent Adapter:** The latent adapter has been integrated into the Semantic-Aware Condition Generation module (DiT) to provide a clearer representation of its application and functionality.
> > > 3. **Target Video Frames:** We have replaced the blurry video frames with the ground truth, as these frames serve as the training targets. This change has been explicitly indicated in the figure to eliminate any potential confusion.
> > >
> > > We have incorporated these modifications into Figure 2. We hope these modifications effectively address the reviewer’s concerns and enhance the overall readability and comprehensibility of Figure 2. Thank you once again for your insightful feedback!

---

> > > > ### Comment · Reviewer_gQNT · 2024-11-25
> > > >
> > > > Thank the authors for the instant and informative response. Now the updated figures are much clearer. I'm having several minor suggestions:
> > > >
> > > >  - Since the DiT model is executed following the auto-regressively generated tokens, it's better to have ellipsis to the right end if Fig. 1, indicating that depending on the target video length, more inference segments could be involved.
> > > >
> > > >  - Out of the same reason for long video sequence generation, when comparing the inference efficiency (Sec. A.2 and Tab. 4), it would benefit to compare the total time or float point operations given a certain long sequence involving more than one inference segment (or a multiplier varialbe $n$ is also fine). Currently there is only one forward pass of diffusion models included, which cannot reflect the actual full inference process.
> > > >
> > > >  - The hierarchical generation of cascaded AR-diffusion is a novel framework. Is there any related work with similar prototype or inspiring you? I see that Sec. 3.2 has been enriched while it would help if more discussion could be provided from this perspective (e.g. how are the sliding window or diffusion-over-diffusion etc. methods relevant and different from your proposed one; why does your proposed frame condition connection outperform previous key frame injection approach; etc).
> > > >
> > > > Overall I'd like to move to a more solid leaning toward acceptance of this work.

---

> > > > > ### Author Response · Authors · 2024-11-26
> > > > >
> > > > > We sincerely appreciate the reviewer for their valuable suggestions and feedback on our responses. We will address each comment individually below.
> > > > >
> > > > > **Q1:** Since the DiT model is executed following the auto-regressively generated tokens, it's better to have ellipsis to the right end if Fig. 1, indicating that depending on the target video length, more inference segments could be involved.
> > > > >
> > > > > **Response:** We thank the reviewer for the insightful comment!
> > > > >
> > > > > In the revised version of the manuscript, an ellipsis has been added at the right end of Figure 1.
> > > > >
> > > > > Thank you once more for your meticulous and perceptive suggestions, which have not only highlighted the finer details but also contributed to elevating the quality of our paper.
> > > > >
> > > > >
> > > > >
> > > > > **Q2:** Out of the same reason for long video sequence generation, when comparing the inference efficiency (Sec. A.2 and Tab. 4), it would benefit to compare the total time or float point operations given a certain long sequence involving more than one inference segment (or a multiplier n varialbe is also fine). Currently there is only one forward pass of diffusion models included, which cannot reflect the actual full inference process.
> > > > >
> > > > > **Response:** We thank the reviewer for the insightful comment!
> > > > >
> > > > > We agree with the reviewer's points that 1) the inference of a single video segment does not fully capture the actual scenario of a long video generation; and 2) the analysis of long video generation can indeed further highlight the strengths of our proposed solution.
> > > > >
> > > > > In addition to reporting the inference speed and FLOPs for a 68-frame 512×512 video, we have now included these two metrics for a 578-frame 512×512 video as well. 578 can be expressed as 68 × 11 - 17 × 10. Here, 11 represents the number of times the DiT model generates 68-frame video segments, while 17 signifies the number of frames in the conditioned video. Additionally, 10 indicates the number of times the DiT model generates videos under specific conditions. As evidenced by the revised Table 4, ARLON demonstrates a **49%** relative enhancement in inference speed and a **64%** relative decrease in computational FLOPs when processing a 578-frame 512×512 video.
> > > > >
> > > > > We are grateful to the reviewer for the suggestions in further highlighting our model's capabilities.

---

> ### Author Response · Authors · 2024-11-26
>
> **Q3:** The hierarchical generation of cascaded AR-diffusion is a novel framework. Is there any related work with similar prototype or inspiring you? I see that Sec. 3.2 has been enriched while it would help if more discussion could be provided from this perspective (e.g. how are the sliding window or diffusion-over-diffusion etc. methods relevant and different from your proposed one; why does your proposed frame condition connection outperform previous key frame injection approach; etc).
>
> **Response:** We thank the reviewer for the insightful comment!
>
> We were inspired by advancements in the speech generation domain, with a notable example being VALL-E [1]. This work incorporates an autoregressive model to generate the first layer codes of Encodec [2], followed by a non-autoregressive model to predict the subsequent layers of codes. For ARLON, it first generates long-term, coarse-grained discrete visual units (AR codes) autoregressively using a decoder-only Transformer. These discrete AR codes are then segmented and sequentially fed into the DiT model to autoregressively generate high-quality video segments.
>
> At the same time, we would like to illustrate the difference between these two models, even ignoring the difference in modality. **The information percentage of AR codes varies significantly between ARLON and VALL-E**. In ARLON, AR codes are characterized by a coarse granularity, primarily due to the highly compact nature of the latent VQ-VAE. Conversely, in VALL-E, the first layer codes are densely packed with the majority of the information, a result of the Residual-VQ mechanism employed by Encodec. Therefore, within the ARLON framework, the DiT model plays a more central role in video generation. In contrast, within the VALL-E system, the AR model is given priority.
>
> Moreover, We would like to elaborate on the relevance and differences of our proposed method compared to existing techniques. Firstly, while sliding window methods ensure consistency between adjacent segments, they struggle to capture long-range dependencies in videos. In contrast, keyframe injection methods often require maintaining a similar appearance to the keyframes; however, this does not guarantee continuity in the action scenes, as relying solely on a single image for constraints can be limiting. As for diffusion-over-diffusion methods, they typically generate keyframes first and then stitch them together, which can lead to a loss of continuity in actions over time, making them more suitable for generating cartoon-style sequences. Our proposed method effectively integrates an autoregressive model for long-term coherence (AR) with a diffusion-based DiT model for short-term continuity (DiT), overcoming the limitations of existing techniques such as sliding window and diffusion-over-diffusion methods. **This approach ensures video integrity and detail coherence over extended periods without repetition.**
> We have detailed the relevance and differences of our proposed method compared to existing techniques in the related work.
>
> [1]. Wang C, Chen S, Wu Y, et al. Neural codec language models are zero-shot text to speech synthesizers[J]. arXiv preprint arXiv:2301.02111, 2023.
>
> [2]. Défossez A, Copet J, Synnaeve G, et al. High fidelity neural audio compression[J]. arXiv preprint arXiv:2210.13438, 2022.

---

> ### Author Response · Authors · 2024-11-26
>
> Dear Reviewer gQNT,
>
> We sincerely thank you for your thoughtful feedback and for taking the time to review our response and update your evaluation. Your insights and suggestions have been invaluable in enhancing our manuscript, and we deeply appreciate your engagement. We are also grateful for your recognition of ARLON's novelty and effectiveness, as well as your increased rating.
>
> Best regards,
>
> Authors

---

### Official Review · Reviewer_Rwnx · 2024-11-04

**Soundness:** 3
**Presentation:** 3
**Contribution:** 3
**Rating:** 6
**Confidence:** 2

**Summary:**

This paper proposes a long video synthesis pipeline, ARLON. The main idea of this paper is to combine DiT with autoregressive transformers that provide long-range temporal information. To bridge the DiT and the AR transformer, the pipeline novelly adopts 1) a Latent VQ-VAE to enable the AR model to learn on and the DiT to learn on different latent spaces, reducing learning complexity and allowing the AR model to manage coarse temporal dependencies; 2) an adaptive norm-based semantic injection module to guide the DiT using AR generated tokens. Novel training strategies of coarser visual latent tokens for DiT and uncertainty sampling are also proposed to make the training process more stable and generalizable.

Results are compared with current t2v models over VBench and Vbench-long and achieve notable improvements especially on long video generation.

**Strengths:**

- The motivation for using the AR model to provide semantic guidance is clear and nice.
- It is a very good extension of existing architectures.
- Good presentation.
- Good qualitative and quantitive results.

**Weaknesses:**

I overall like this paper, but there are several points for improvement.

- No ablation on the impact of model structure and training data size.
- No discussion on failure cases and limitations.
- There might be some missing references such as nuwa-XL and Phenaki. GAN-based long video generation might also be related.

*[1] NUWA-XL: Diffusion over Diffusion for eXtremely Long Video Generation*

*[2] Phenaki: Variable Length Video Generation from Open Domain Textual Descriptions*

I am not an expert in this field of training large video generation models. I will adjust the final score with other reviewers' comments and also based on the response from the author.

**Questions:**

- How many seconds can the models generate for the longest videos and how is the performance? For How many seconds do the longest videos that your method generates can last? In my understanding, this is the key advantage of the hierarchical generation framework.
- The main limitation of this work seems to be the huge computational cost of training, but the related information (type and number of GPU, training time) is not provided. It would be nice to know this information.

**Details Of Ethics Concerns:**

The training datasets (Openvid, ChronoMagic-ProH, and OpenSora-plan) used in this paper are all open-sourced as listed in the paper. and the authors might not choose to open-source the models. However, text-to-video models can generate harmful and troublesome content is a broad concern, the discussion on this problem is needed.

---

> ### Author Response · Authors · 2024-11-22
> **Response to Reviewer Rwnx - Part 1**
>
> We would like to express our gratitude to the reviewer for acknowledging the motivation and contributions of our ARLON, the promising qualitative and quantitative results, and the quality of our paper's writing. We would also like to thank the reviewer for the constructive suggestions and comments which will be responded to one-by-one below.
>
>
>
> **W1.** No ablation on the impact of model structure and training data size.
>
> **Response**: We thank the reviewer for the insightful comments!
>
> We would like to clarify that the analysis of the model structure has already been provided in the second point of Section 4.3. Specifically, we compare the performance of the **semantic injection module across various structural configurations**, including ControlNet, MLP adapter and adaptive norm.
>
> Additionally, we would like to present an analysis of the impact of varying training data sizes.
>
> | Dataset                       | **Subject consistency** | **Background consistency** | **Motion smoothness** | **Dynamic degree** | **Aesthetic quality** | **Imaging Quality** |
> | ----------------------------- | ----------------------- | -------------------------- | --------------------- | ------------------ | --------------------- | ------------------- |
> | Openvid-1M                    | 95.02                   | 96.35                      | 98.16                 | 30.00              | 52.34                 | 59.15               |
> | Openvid-HQ 0.4M               | 97.78                   | 97.83                      | 99.25                 | 30.00              | 55.42                 | 64.11               |
> | Openvid-HQ 0.4M + Mixkit 0.3M | 97.39                   | 97.55                      | 99.24                 | 34.00              | 56.90                 | 65.33               |
>
> Firstly, when comparing our model's performance on the OpenVid 1M dataset with that on OpenVid-HQ (which contains higher quality videos, totaling 0.4M), we observed a marked improvement in our model's performance on OpenVid-HQ. **This indicates that the quality of the data plays a crucial role in the task of video generation.**
>
> Furthermore, when we combined the OpenVid-HQ and Mixkit (the quality of videos is also high) datasets as our training set (approximately 0.7M), improvements in both quality and dynamic degree are obtained. This suggests that in the context of video generation, **prioritizing high-quality videos while also utilizing a larger dataset** can effectively enhance the overall quality of generated videos.
>
> We have incorporated these results into the Appendix, which can be found in Table 5. We greatly appreciate your valuable feedback and suggestions.
>
>
>
> **W2.** No discussion on failure cases and limitations.
>
> **Response:** We thank the reviewer for the insightful comments!
>
> We have included the failure cases in the Appendix and provided explanations for why these cases failed, along with an analysis of potential future directions for exploration. **The details can be found in Appendix Section A4 and Figure 16 of the revised manuscript.**
>
> In addition, **Limitations and Broader Impact** are also provided in the Appendix, which can also be found in the response to **Ethics Concerns.**
>
>
>
> **W3.** There might be some missing references such as nuwa-XL and Phenaki. GAN-based long video generation might also be related.
>
> **Response:** We thank the reviewer for the insightful comments!
>
> We have revised the related work section to include these references including NUWA-XL, Phenaki and GAN-based long video generation works, and discuss their relevance to our approach.
>
> Please find them in Section 3.2 of the updated manuscript (Highlighted in blue).

---

> ### Author Response · Authors · 2024-11-22
> **Response to Reviewer Rwnx - Part 2**
>
> **Q1.** How many seconds can the models generate for the longest videos and how is the performance? For How many seconds do the longest videos that your method generates can last? In my understanding, this is the key advantage of the hierarchical generation framework.
>
> **Response:** We thank the reviewer for the insightful questions.
>
> For our model, ARLON, as depicted in Figure 1 of the paper, all coarse AR codes are generated in **a single inference pass**, providing **coarse spatial and long-range temporal** information. This effectively guides the DiT model to autoregressively produce high-quality videos with rich dynamic motion. In comparison to previous methods, each segment generated by ARLON exhibits notable consistency, leading to an outstanding performance in terms of both the **quality and consistency** of the final long video. we agree that this is the one of key advantages of ARLON. The longest duration of our training video clips for AR model is about 60 seconds, therefore the quality of **generated videos by ARLON can last at least 60 seconds**.
>
> Although employing autoregressive approaches for long video generation using DiT models—where successive video segments are generated based on the last frames of the previous segment— theoretically allows for the generation of videos of almost infinite length. However, the inherent computational constraints limit the length of these conditioned segments, thereby **restricting the historical context** available for the generation of each new segment. Moreover, when identical text prompts are used, the generated short video segments frequently exhibit identical content, heightening the risk of **repetition throughout the entire long video**. Additionally, the process of autoregressive generation inevitably leads to the **accumulation of errors**. Given these three critical points, the quality of videos generated using solely autoregressive DiT models is notably poor.
>
>
>
> **Q2.** The main limitation of this work seems to be the huge computational cost of training, but the related information (type and number of GPU, training time) is not provided. It would be nice to know this information.
>
> **Response:** We thank the reviewer for the insightful comments!
>
> We would like to clarify that the computational cost of training our model is not huge compared to the baseline OpenSora-V1.2. This is because we initialized the parameters of the DiT model (1.2 B) from OpenSora-V1.2 and subsequently froze them. Consequently, only an additional 192 M + 92 M + 30 M parameters require training (for more details, please refer to the **Summary response**).
>
> For your interest in the computational aspects of our work, we would like to provide the details. All experiments were conducted on NVIDIA A100 40G GPUs. Specifically, the AR model requires **one day of training on 8 NVIDIA A100 40G GPUs**, while the DIT model takes **two days of training on 32 NVIDIA A100 40G GPUs** (For reference, the training of OpenSora-v1.1 requires approximately 9 days on 64 H800 GPUs). Therefore, our method demonstrates a lower computational cost. We hope this information addresses your concerns.

---

> ### Author Response · Authors · 2024-11-22
> **Response to Reviewer Rwnx - Part 3**
>
> **Ethics Concerns:** Text-to-video models can generate harmful and troublesome content is a broad concern, the discussion on this problem is needed.
>
> **Response**: We thank the reviewer for the insightful comments!
>
> We would like to provide the **Limitations and Broader Impact.**
>
> **Limitations**
>
> Although ARLON achieves state-of-the-art performance in long video generation, it also exhibits some specific constraints. First, ARLON is built upon OpenSora-V1.2, which potentially caps the upper limit of video quality. Nonetheless, this limitation can be mitigated by substituting the DiT model with more advanced alternatives, such as CogVideoX-5B or MovieGen. Second, if we aim to train ARLON at 2K resolution, the sequence length of AR codes will become excessively long, making both training and inference impractical. Viable solutions involve employing a higher compression ratio in VQ-VAE, or selectively retaining essential information while disregarding irrelevant details. Additionally, for the AR model, parallel prediction emerges as an alternative approach.  Our future research endeavors will delve into addressing these issues.
>
> **Broader Impact**
>
> Synthetic video generation is a powerful technology that can be misused to create fake videos or videos containing harmful and troublesome content, hence it is important to limit and safely deploy these models. From a safety perspective, we emphasize that the training data of ARLON are all open-sourced, and we do not add any new restrictions nor relax any existing ones to OpenSora-V1.2.  If you suspect that ARLON is being used in a manner that is abusive or illegal or infringes on your rights or the rights of other people, you can report it to us.
>
> We also add the **Limitations and Broader Impact** into the Appendix, please find them in A.4 and A.5.

---

> ### Author Response · Authors · 2024-11-26
>
> Dear Reviewer Rwnx,
>
> We sincerely appreciate the valuable feedback provided you have provided. We have taken great care to address all the concerns in detail. As we near the conclusion of the discussion phase, we genuinely value your insights and would be grateful for any further feedback you may have. We hope that our responses meet your expectations and remain open to addressing any additional questions you might have. Thank you once again for your time and consideration.
>
> Best regards,
>
> Authors

---

> > ### Author Response · Authors · 2024-12-01
> >
> > Dear Reviewer Rwnx,
> >
> > We would like to express our sincere gratitude for your valuable feedback on our paper. We have carefully considered your comments and have made the necessary revisions and improvements based on your suggestions.
> >
> > As the discussion deadline is approaching on December 2, we kindly request that you review our responses and provide any additional feedback at your earliest convenience. Your insights are crucial to us, and we hope to address any remaining concerns promptly.
> >
> > Thank you once again for your time and effort in reviewing our work. We greatly appreciate your support and look forward to your feedback.
> >
> > Best regards,
> >
> > Authors

---

### Official Review · Reviewer_N9At · 2024-11-07

**Soundness:** 2
**Presentation:** 3
**Contribution:** 3
**Rating:** 5
**Confidence:** 2

**Summary:**

The manuscript introduces ARLON, a text-to-video framework that efficiently generates high-quality, dynamic, and temporally consistent long videos. By combining Autoregressive models with Diffusion Transformers, ARLON employs innovations like VQ-VAE for token compression, an adaptive semantic injection module, and an uncertainty sampling module to enhance efficiency and noise tolerance. It reduces denoising steps and outperforms OpenSora-V1.2 in both quality and speed, achieving state-of-the-art performance.

**Strengths:**

1. The integration of Autoregressive models with Diffusion Transformers and innovations like VQ-VAE for token compression, adaptive semantic injection, and uncertainty sampling show originality in addressing long video generation challenges.
2. The generated video spans 600 frames, making it relatively long.

**Weaknesses:**

1. In Table 2, why does StreamingT2V have a higher Dynamic Degree score (85.64) compared to ARLON (50.42)?
2. The paper lacks a detailed comparison of the number of parameters in ARLON versus baseline models.
3. There is no analysis of ARLON's memory footprint during training and inference, which would clarify its computational efficiency relative to models like OpenSora-V1.2.

**Questions:**

Please refer to the weakness part.

---

> ### Author Response · Authors · 2024-11-22
> **Response to Reviewer N9At - Part 1**
>
> We would like to thank the reviewer for highlighting the novelty of our proposed methods and the capability to generate long videos. We would also like to thank the reviewer for the constructive suggestions and comments which will be responded to one-by-one below.
>
> **W1.** In Table 2, why does StreamingT2V have a higher Dynamic Degree score (85.64) compared to ARLON (50.42)?
>
> **Response:** We appreciate the reviewer's insightful question.
>
> As illustrated in Figures 4 (Figure 5 in the original manuscript) and 14, and further exemplified in the "Long Video Results" section of the demo page, the videos generated by StreamingT2V exhibit notable **fluctuations in object motion and background over time**. These include instances of sudden object transitions, abrupt scene alterations, and screen disruptions. These characteristics collectively contribute to a high Dynamic Degree score for StreamingT2V. It should be noted that this dynamism comes **at the expense of subject consistency, background consistency, and aesthetic quality**. Specifically, StreamingT2V scores lower in these aspects, resulting in videos that are, subjectively, less enjoyable to watch.
>
> We want to highlight that **ARLON's primary strength lies in its ability to generate long videos with remarkable consistency and quality**. By leveraging the AR model, ARLON achieves top scores in subject consistency, background consistency, and aesthetic quality, while also delivering impressive dynamic scores in long video generation, as shown **in Table 1** (Table 2 in the original manuscript)**.**
>
>
>
> **W2.** The paper lacks a detailed comparison of the number of parameters in ARLON versus baseline models.
>
> **Response:** We thank the reviewer for the insightful comments!
>
> We would like to give the number of parameters for each component in ARLON compared to OpenSora-V1.2.
>
> | **Method**             | **OpenSora-V1.2 (baseline)** | **ARLON (ours)**                    |
> | ---------------------- | ---------------------------- | ----------------------------------- |
> | Param. (AR)            | -                            | **192M**                            |
> | Param. (DiT)           | 1.2B                         | **92M (trainable)** + 1.2B (frozen) |
> | Param. (3D VAE)        | 384M                         | 384M                                |
> | Param. (latent VQ-VAE) | -                            | **30M**                             |
>
> As illustrated in the Table, the increase in the number of parameters (**192M + 92M + 30M**) of our method is  minimal compared to the **1.2B** parameters of the baseline, OpenSora-V1.2. This is primarily due to our adoption of an efficiency adapter approach during training, which enables us to introduce fewer parameters while preserving performance. Additionally, the latent VQ-VAE operates within a compressed latent space, and the AR model is trained to generate highly quantized tokens, both of which significantly contribute to the minimized parameter requirements.
>
> We have incorporated these details into the Appendix, which can be found in Table 4. We greatly appreciate your valuable feedback and suggestions.

---

> ### Author Response · Authors · 2024-11-22
> **Response to Reviewer N9At - Part 2**
>
> **W3.** There is no analysis of ARLON's memory footprint during training and inference, which would clarify its computational efficiency relative to models like OpenSora-V1.2.
>
> **Response:** We thank the reviewer for the insightful comments!
>
> We would like to provide the detailed memory requirements for both training and inference of ARLON:
>
> Caption: The batch size for training both the AR and DiT models is set to 2x68-frame 512x512 video clips, and the inference is evaluated by generating a 68-frame 512x512 video. Both the training and inference processes are executed on a single 40G A100 GPU.
>
> | **Method**       | **OpenSora-V1.2 (baseline)** | **ARLON (ours)**                      |
> | ---------------- | ---------------------------- | ------------------------------------- |
> | Training Memory  | 36007 MB                     | 7701 MB (AR); 36815 MB (DiT)          |
> | Inference Memory | 24063 MB                     | 2269 MB (AR); 25215 MB (DiT)          |
> | Inference speed  | 47.3 s                       | 5.7s (AR) + 18.9s (DiT)               |
> | Inference FLOPs  | 42626G × 30 (steps)          | 200G (AR) + 46461G × 10 (steps) (DiT) |
>
> From the Table, it is evident that our method does not require significant additional memory and computational resources compared to the baseline model. Specifically, during the training and inference phases, there are **2.2% and 4.8%** relative increases respectively (the AR and DiT models can be trained independently).
>
> Conversely, by leveraging the AR code as an efficient initialization, our model is capable of generating high-quality videos with significantly fewer steps. Consequently, our inference time and total FLOPs are superior to those of the baseline, thereby significantly accelerating the denoising process. Specifically, our model achieves a **48%** relative improvement in inference speed and a **64%** relative reduction in computational FLOPs. For further results of long video (578-frame) generation, please refer to the general response titled "Model details".
>
> We have incorporated these details into the Appendix, which can be found in Table 4. We greatly appreciate your valuable feedback and suggestions.

---

> ### Author Response · Authors · 2024-11-26
>
> Dear Reviewer N9At,
>
> Thank you for your thorough review of our paper. In our previous response and the revised manuscript, we conducted additional experiments and provided detailed explanations addressing your questions and concerns. As we near the end of the author-reviewer discussion phase, we kindly ask you to review our revised manuscript and our responses, and consider reevaluating our work if we have satisfactorily addressed all your concerns. If you have any further questions, please feel free to reach out, and we would be delighted to offer any additional clarification.
>
> Best regards,
>
> Authors

---

> > ### Author Response · Authors · 2024-12-01
> >
> > Dear Reviewer N9At,
> >
> > We would like to express our sincere gratitude for your valuable feedback on our paper. We have carefully considered your comments and have made the necessary revisions and improvements based on your suggestions.
> >
> > As the discussion deadline is approaching on December 2, we kindly request that you review our responses and provide any additional feedback at your earliest convenience. Your insights are crucial to us, and we hope to address any remaining concerns promptly.
> >
> > Thank you once again for your time and effort in reviewing our work. We greatly appreciate your support and look forward to your feedback.
> >
> > Best regards,
> >
> > Authors

---

### Author Response · Authors · 2024-11-22
**Model Details**

Caption: The batch size for training both the AR and DiT models is set to 2×68-frame 512×512 video clips, and the inference is evaluated by generating a 68-frame (68f) or 578-frame (578f) 512×512 video. Both the training and inference processes are executed on a single 40G A100 GPU.

| **Method**             | **OpenSora1.2 (baseline)**      | **ARLON (ours)**                                 |
| ---------------------- | ------------------------------- | ------------------------------------------------ |
| Param. (AR)            | -                               | 192M                                             |
| Param. (DIT)           | 1.2B                            | 92M (trainable) + 1.2B (frozen)                  |
| Param. (3D VAE)        | 384M                            | 384M                                             |
| Param. (latent VAE)    | -                               | 30M                                              |
| Traning Memory         | 36007M                          | 7701M (AR) ; 36815MB (DiT)                       |
| Inference Memory       | 24063M                          | 2269M (AR) ; 25215M (DiT)                        |
| Inference speed (68f)  | 47.3 s                          | 5.7s (AR)+18.9s (DiT)                            |
| Inference speed (578f) | 47.3 × 11 s                     | 57.2s (AR) + (18.9 × 11) s (DiT)                 |
| Inference FLOPs (68f)  | 42626G × 30 (step)              | 200G (AR) + 46461G×10 (step) (DiT)               |
| Inference FLOPs (578f) | 42626G × 30 (step) × 11 (times) | 1547G (AR) + 46461G×10 (step) × 11 (times) (DiT) |

From the results of the table, we can observe that:

1. the increase in the number of parameters (**192M + 92M + 30M**) of ARLON is minimal compared to the **1.2B** parameters of the baseline, OpenSora-V1.2. This is primarily due to our adoption of an efficiency adapter approach during training, which enables us to introduce fewer parameters while preserving performance. Additionally, the latent VQ-VAE operates within a compressed latent space, and the AR model is training and generates highly quantized tokens, both of which significantly contribute to the minimized parameter requirements.
2. it is evident that our method does not require significant additional memory and computational resources compared to the baseline model. Specifically, during the training and inference phases, there are **2.2% and 4.8%** relative increases respectively (the AR and DiT models can be trained independently).
3. Conversely, by leveraging the AR code as an efficient initialization, our model is capable of generating high-quality videos with significantly fewer steps. Consequently, our inference time and total FLOPs are superior to those of the baseline, thereby significantly accelerating the denoising process. Specifically, our model achieves a **48-49%** relative improvement in inference speed and a **64%** relative reduction in computational FLOPs for 68-frame or 578-frame video generation (**578 can be expressed as 68 × 11 - 17 × 10**. Here, 11 represents the number of times the DiT model generates 68-frame video segments, while 17 signifies the number of frames in the conditioned video. Additionally, 10 indicates the number of times the DiT model generates videos under specific conditions).

---

### Author Response · Authors · 2024-11-22
**Summary**

We would first like to express our gratitude to Program Chairs, Senior Area Chairs, and Area Chairs for their efforts, as well as to the dedicated reviewers for their insightful comments on our paper.

Additionally, we appreciate the reviewers for highlighting the **strengths** of our work:

1. Novel framework, that seamlessly combines the strengths of autoregressive Transformers and Diffusion Transformers (DiT).
2. Novel technical designs, including a latent VQ-VAE, adaptive semantic injection and noise-resilient strategies.
3. State-of-the-art long video generation performance, and the faster generation process with comparable performance.
4. The paper is written and presented clearly and easy to follow.

On the other hand, based on the constructive feedback, suggestions and comments from reviewers, we will make the following revisions to our paper:

1. We would like to emphasize that ARLON's primary strength lies in its ability to generate **long videos with remarkable consistency and quality**, while also delivering **impressive dynamic scores.** To more effectively showcase our contributions, we have adjusted the sequence of our presentation: we now begin with the results of the long videos, followed by those of the short videos, thereby **swapping the original order of Tables 1 and 2**, as well as **that of Figures 4 and 5.**
2. In Appendix, we have added **i)** comprehensive implemental details, encompassing the number of parameters, memory requirements for both training and inference,  inference speed, and FLOPS, for both the baseline OpenSora-V1.2 and our advanced ARLON models; **ii)** the failure cases and analysis for these cases; **iii)** Limitations and Broader Impact; **iv)** results and analysis of various size of training data for DiT model.
3. In related work, We have added NUWA-XL and Phenaki, and GAN-based long video generations, and discussed their relevance to our approach. In addition, all of them have been added into Reference.
4. For Table 2 (Table 1 in the original manuscript), we have divided it into two sections. The lower section presents the results of ARLON and OpenSora-V1.2, with our superior results highlighted with specific improvements. The upper section, meanwhile, displays the results of other text-to-video models. In addition, we have added the sentence "the higher scores of metrics indicate better performance." to the table captions.
5. For Table 3, we have changed the column name "Compress Ratio" to "Compress Ratio in DiT", and added the description "The compression ratio of the latent VQ-VAE for AR model is 4x8x8" into the table caption.
6. In Section 4.3, we have revised the content to avoid any confusion for the readers ("which could simulate the errors" -> "which could make the DiT model tolerate the errors").

Once again, we sincerely thank you for your invaluable input and the time you have dedicated to reviewing our paper. Your constructive feedback and insightful suggestions have been instrumental in enhancing the quality of our research.

---

> ### Author Response · Authors · 2024-11-23
>
> we wanted to reach out to confirm that all your concerns have been adequately addressed. Should you have any further questions or require additional clarifications, please do not hesitate to discuss with us at your earliest convenience.

---

### Meta-Review · Area_Chair_BjwV · 2024-12-17

**Metareview:**

This paper explores the generation of long videos guided by an autoregressive language model (LM) to produce conditions for a diffusion transformer (DiT) model. It introduces new techniques to bridge the gap between the LM and the DiT, including a new VQ-VAE that quantizes the DiT model’s input into visual tokens and tolerates noise robustness.

The reviewers generally acknowledge the paper’s strengths, such as its clear motivation, innovative techniques, and empirical results for generating long video sequences. While three out of four reviewers favor accepting the submission, one reviewer leans against it.

Unfortunately, the opposing reviewer (N9At) did not respond to requests from either the authors or the AC to engage with the authors' rebuttal.

The AC reviewed both the opposing review and the authors’ response and believes the raised concerns may have been addressed. Of the three main questions raised by Reviewer N9At, two were requests for clarification, to which the authors provided answers. The third question, which sought additional explanation of a performance difference to the baseline, is viewed by the AC as not a significant concern, and the authors provided a reasonable response.

**Additional Comments On Reviewer Discussion:**

Only one reviewer responded to the rebuttal, indicating that their concerns were resolved. The opposing reviewer (N9At) did not provide a response nor answered the AC's request to do so. The AC read reviewed the feedback and the authors' rebuttal.

---

### Decision · Program_Chairs · 2025-01-22

Accept (Poster)